# A transcription factor ensemble orchestrates bundle sheath expression in rice

Lei Hua ✉, Na Wang, Susan Stanley, Ruth M. Donald ⓘ, Satish Kumar Eeda, Kumari Billakurthi ⓘ, Ana Rita Borba & Julian M. Hibberd ⓘ ✉

C$_4$ photosynthesis has evolved in over sixty plant lineages and improves photosynthetic efficiency by ~50%. One unifying character of C$_4$ plants is photosynthetic activation of a compartment such as the bundle sheath, but gene regulatory networks controlling this cell type are poorly understood. In Arabidopsis, a bipartite MYC-MYB transcription factor module restricts gene expression to these cells, but in grasses the regulatory logic allowing bundle sheath gene expression has not been defined. Using the global staple and C$_3$ crop rice, we find that the *SULFITE REDUCTASE* promoter is sufficient for strong bundle sheath expression. This promoter encodes an intricate *cis*-regulatory logic with multiple activators and repressors acting combinatorially. Within this landscape we identify a distal *cis*-regulatory module (CRM) activated by an ensemble of transcription factors from the WRKY, G2-like, MYB-related, DOF, IDD and bZIP families. This module is necessary and sufficient to pattern gene expression to the rice bundle sheath. Oligomerisation of the CRM and fusion to core promoters containing Y-patches allow activity to be increased 220-fold. This CRM generates bundle sheath-specific expression in *Arabidopsis* indicating deep conservation in function between monocotyledons and dicotyledons. In summary, we identify an ancient, short, and tuneable CRM patterning expression to the bundle sheath that we anticipate will be useful for engineering this cell type in various crop species.

In plants and animals significant progress has been made in understanding transcription factor networks responsible for the specification of particular cell types. In animals, for example, homeobox transcription factors define the body plan of an embryo[1,2], and cardiac cell fate is specified by five transcription factors comprising Pnr and Doc that act as anchors for dTCF, pMad, and Tin[3]. In plants the INDE-TERMINATE DOMAIN (IDD) transcription factors work together with SCARECROW and SHORTROOT to specify endodermal formation in the root[4,5], PHLOEM EARLY (PEAR) and VASCULAR-RELATED NAC DOMAIN (VND) transcription factors permit production of phloem and xylem vessel respectively[6,7], and basic helix–loop–helix (bHLH) transcription factors determine differentiation of guard cells[8–11]. Moreover, transcription factor networks that integrate processes as diverse as responses to external factors such as pathogens and abiotic stresses[12,13], or internal events associated with the circadian clock[14,15] and hormone signalling[16,17] have also been identified. Transcription factor activity is decoded by short *cis*-acting DNA sequences known as *cis*-regulatory elements. The binding of multiple transcription factors to a *cis*-regulatory module (CRM) thus controls transcription and the spatiotemporal patterning of gene expression by boosting or suppressing gene activity[18]. For example, the Block C enhancer interacts with the core promoter to activate expression of *FLOWERING LOCUS T* in long days[19,20], and a distant upstream enhancer controls expression of the *TEOSINTE BRANCHED1* locus in maize responsible for morphological differences compared with the wild ancestor teosinte[21,22]. In contrast to the above examples, transcription factors and cognate *cis*-elements responsible for the operation of cell types in grasses once specified, have not been defined[23,24].

Department of Plant Sciences, University of Cambridge, Downing Street, Cambridge, United Kingdom. ✉e-mail: lh556@cam.ac.uk; jmh65@cam.ac.uk

Given the increased specialisation of plant organs since the colonisation of land this lack of understanding of gene regulatory networks controlling cell-specific gene expression is striking. For example, in the liverwort *Marchantia polymorpha* the photosynthetic thallus contains seven cell types[25], while leaves of *Oryza sativa* (rice) and *Arabidopsis thaliana* possess at least fifteen and seventeen populations of cells as defined by single-cell sequencing, respectively[26]. In leaves of these angiosperms, particular cell types are specialised for photosynthesis and so activation of photosynthesis gene expression is highly responsive to light[27] One such cell type is the bundle sheath, and while these cells carry out photosynthesis they are also specialised to allow water transport from veins to mesophyll, as well as sulphur assimilation and nitrate reduction[28–30]. And, strikingly in multiple lineages, the bundle sheath has been dramatically repurposed during evolution to become fully photosynthetic and allow the complex $C_4$ pathway to operate[31].

Compared with the ancestral $C_3$ state, plants that use $C_4$ photosynthesis operate higher light, water, and nitrogen use efficiencies[31–33]. It is estimated that introducing the $C_4$ pathway into $C_3$ rice would allow a 50% increase in yield[33,34], but it requires multiple photosynthesis genes to be expressed in the bundle sheath, including enzymes that decarboxylate $C_4$ acids to release $CO_2$ around RuBisCO, organic acid transporters, components of the Calvin-Benson-Bassham cycle, RuBisCO activase, and enzymes of starch biosynthesis[35–37]. In summary, although the bundle sheath is found in all angiosperms and is associated with multiple processes fundamental to leaf function, the molecular mechanisms responsible for directing expression to this cell type, including in global staple crops, remain undefined. We therefore studied the bundle sheath to better understand the complexity of gene regulatory networks that operate to maintain the function of a cell type once it has been specified. Rice was chosen as it is a global crop, and identifying how it patterns gene expression in the bundle sheath could facilitate engineering of this cell type.

We hypothesised that analysis of endogenous patterns of gene expression in the rice bundle sheath would allow us to identify a strong and early-acting promoter for this cell type. Once such a promoter was identified, we also hypothesised that it could be used to initiate an understanding of the *cis*-regulatory logic that allows gene expression to be patterned to this cell type in grasses. We tested twenty-five promoters from rice genes that transcriptome sequencing indicated were highly expressed in these cells. Of these, four specified preferential expression in the bundle sheath, and one derived from the *SULPHITE REDUCTASE* (*SiR*) gene (nucleotides −2571 to +42 relative to translational start site) generated strong bundle sheath expression from plastochron 3 leaves onwards. Truncation analysis showed that bundle sheath expression pattern from the *SiR* promoter is mediated by a short distal CRM and a pyrimidine patch (Y-patch) in the core promoter. This bundle sheath module is cryptic until other CRMs acting to both constitutively activate and repress expression in mesophyll cells are removed. The CRM is composed of a sextet of *cis*-elements recognised by their cognate transcription factors from the WRKY, G2-like, MYB-related, DOF, IDD, and bZIP families. These transcription factors act synergistically and are sufficient to drive expression of the strong bundle sheath *SiR* promoter.

## Results

### The *SiR* promoter directs expression to the rice bundle sheath

To identify sequences allowing robust expression in rice bundle sheath cells, we initially used data derived from laser capture microdissection of bundle sheath strands (comprising bundle sheath, xylem, and phloem) and mesophyll cells from mature leaves. To identify regulatory regions, upstream promoter sequences and where relevant DNase I hypersensitive sites that extended into coding sequence[38] from seven of the most strongly expressed genes in bundle sheath strands were cloned, fused to the β-glucoronidase (GUS) reporter and

transformed into rice (Supplementary Fig. 1a, Supplementary Data 3). Although five of these fusions (*MYELOBLASTOSIS*, *MYB*; *HOMOLOGUE OF E. COLI BOLA*, *bolA*; *GLUTAMINE SYNTHETASE 1*, *GS1*; *STRESS RESPONSEIVE PROTEIN*, *SRP*; *ACYL COA BINDING PROTEIN*, *ARP*) led to GUS accumulation, it was restricted to veins (Supplementary Fig. 1b, c). For the *SULPHATE TRANSPORTER 3;1* and *3;3* (*SULT3;1* and *SULT3;3*) promoters, no staining was observed (Supplementary Fig. 1b, c). The approach of cloning promoters from bundle sheath strands therefore appeared to be more efficient at identifying sequences capable of driving expression in veins. Using an optimised procedure to separate bundle sheath cells from veins[39], we therefore produced transcriptomes from mesophyll, bundle sheath and vascular bundles and identified clusters of genes associated with each cell type[30]. Eighteen of the genes most differentially expressed between bundle sheath and mesophyll, and associated with biological processes such as solute transport, sulphur metabolism, and nitrogen metabolism previously linked to the bundle sheath were selected (Supplementary Fig. 2a). When the promoter from each gene was fused to GUS and transformed into rice, those from *ATP-SULFURYLASE 1B*, *ATPS1b*; *SULPHITE REDUCTASE*, *SiR*; *HIGH ARSENIC CONTENT1.1*, *HAC1.1*; and *FERREDOXIN*, *Fd* were sufficient to generate expression in the bundle sheath (Supplementary Fig. 2b). However, *ATPS1b* and *Fd* also displayed weak activity in the mesophyll, and *HAC1.1* also led to GUS accumulation in epidermal and vascular cells. Thus, only the *SiR* promoter drove strong expression in the bundle sheath and veins with no GUS detected in mesophyll cells (Supplementary Fig. 2b, c). An additional six promoters (*SOLUBLE INORGANIC PYROPHOSPHATASE*, *PPase*; *PLASMA MEMBRANE INTRINSIC PROTEIN1;1*, *OsPIP1;1*; *PLASMA MEMBRANE INTRINSIC PROTEIN1;3*, *OsPIP1;3*; *ACTIN-DEPOLYMERISING FACTOR*, *ADF*; *PEPTIDE TRANSPORTER PTR2*, *PTR2*; *NITRATE REDUCTASE1*, *NIA1*) generated expression in vascular bundles, and eight promoters produced no staining (Supplementary Fig. 2b, c). In summary, most candidate promoters failed to generate expression that was specific to bundle sheath cells, but the region upstream of the rice *SiR* gene was able to do so. We therefore selected the *SiR* promoter for further characterisation.

### The *SiR* promoter drives strong and early expression in bundle sheath cells

Sequence upstream of the *SiR* gene, comprising nucleotides −2571 to +42 relative to the predicted translational start site was sufficient to generate expression in the rice bundle sheath. To allow faster analysis of sequences responsible for this output, we domesticated the sequence by removing four *Bsa*I and *Bpi*I sites such that it was compatible with the modular Golden Gate cloning system. When this modified sequence was placed upstream of the GUS reporter it also generated bundle sheath preferential accumulation (Fig. 1a). Fusion to a nuclear-targeted mTurquoise2 fluorescent protein confirmed that the *SiR* sequence was sufficient to direct expression to bundle sheath cells, and also revealed expression in the longer nuclei of veinal cells (Fig. 1b). Expression from the domesticated and non-domesticated sequences was not different (Fig. 1c). Compared with 0.58 nmol 4-MU/min/mg protein previously reported for the *Zoysia japonica PHOSPHOENOLCARBOXYKINASE (PCK)* promoter[40], activity from the *SiR* promoter was at least 36% higher. Designer Transcription Activator-Like Effector (dTALEs) and cognate Synthetic TALE-Activated Promoters (STAPs) amplify expression and allow multiple transgenes to be driven from a single promoter[41,42]. We therefore tested whether bundle sheath expression mediated by the *SiR* promoter is maintained and strengthened by the dTALE-STAP system. Stable transformants showed bundle sheath-specific expression (Supplementary Fig. 3a, b), and GUS activity was -18-fold higher than that from the endogenous *SiR* promoter (Supplementary Fig. 3c). We conclude that the *SiR* promoter is compatible with the dTALE-STAP system and its activity can be strengthened.

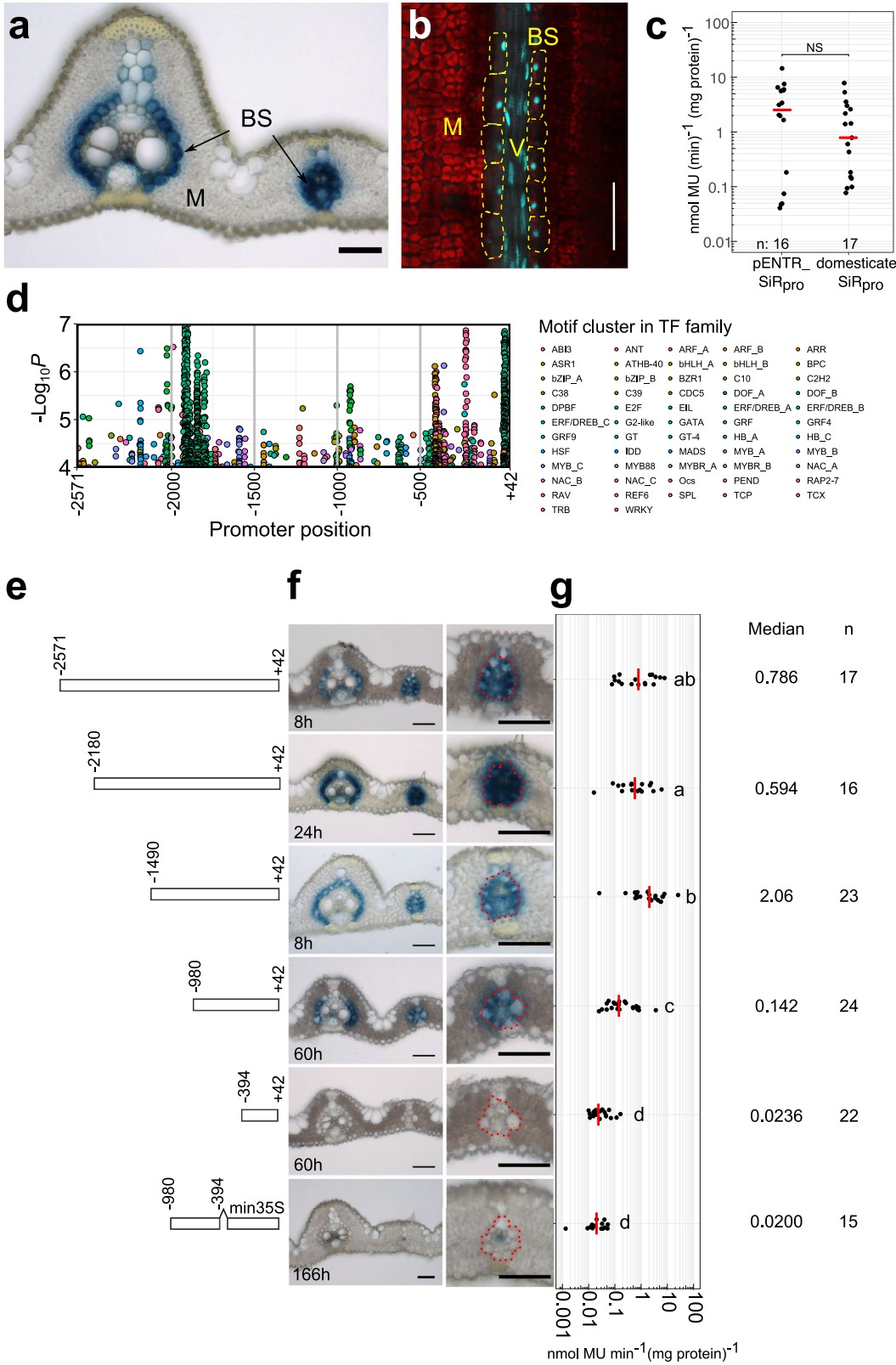

We investigated when promoter activity was first detected during leaf development and discovered that GUS as well as fluorescence from mTurquoise2 were visible in 5–20 mm long fourth leaves at plastochron 3 (Supplementary Fig. 4). This was not the case for the *ZjPCK* promoter even when a dTALE was used to amplify expression (Supplementary Fig. 4). We conclude that the *SiR* promoter initiates expression in the bundle sheath before the *ZjPCK*

promoter, and that it is also able to sustain higher levels of expression in this cell type.

## A distal CRM necessary for expression in the bundle sheath

The *SiR* promoter contains a highly complex *cis* landscape (Fig. 1d) comprising at least 638 predicted motifs from 56 transcription factor families predicted using the Find Individual Motif Occurrences (FIMO)

**Fig. 1 | Nucleotides −980 to −394 of the *SiR* promoter are necessary for bundle sheath expression. a** Domesticated *SiR* promoter generates strong GUS staining in bundle sheath. **b** mTurquoise2 signal driven by the domesticated *SiR* promoter in nuclei of bundle sheath cells (marked by yellow dashed lines) and vein cells in mature leaves, red indicates chlorophyll autofluorescence. **c** The fluorometric 4-methylumbelliferyl-β-D-glucuronide (MUG) assay shows no statistically significant difference between the endogenous and domesticated *SiR* promoter activity, data subjected to two-sided Wilcoxon rank-sum test, n indicates the number of biological replicates (independent T$_0$ transgenic plants), median catalytic rate of GUS indicated with red lines. Source data are provided as a Source Data file. **d** Landscape of transcription factor binding sites in the *SiR* promoter using the Find Individual Motif Occurrences (FIMO) programme. The likelihood of match to 656 plant known transcription factor motifs in the *SiR* promoter is shown by transcription factor

families (Supplementary Data 1), *P* values calculated from the log-likelihood score by the FIMO programme. **e** Schematics showing 5′ truncations. **f** Representative images of leaf cross sections from transgenic lines after GUS staining. Zoomed-in images of lateral veins shown in right panels, the staining duration is displayed in the bottom-left corner, bundle sheath cells highlighted with dashed red line, scale bars = 50 μm. **g** Promoter activity determined by the fluorometric 4-methyl-lumbelliferyl-β-D-glucuronide (MUG) assay. Data subjected to pairwise two-sided Wilcoxon rank-sum test with Benjamini-Hochberg correction for multiple comparison. Lines with differences in activity that were statistically significant (adjusted *P* < 0.05) were labelled with different letters. The median represents the median GUS activity value and is shown as red line, n indicates the number of T$_0$ transgenic plants analysed. Source data are provided as a Source Data file.

tool[43] with position weight matrix from the JASPAR database[44] (Supplementary Data 1). We therefore designed a 5′ truncation series to investigate regions necessary for expression in the bundle sheath (Fig. 1e). Deleting nucleotides −2180 to −1490 and −1490 to −980 led to a statistically significant increase and then reduction in MUG activity respectively but neither truncation abolished preferential accumulation of GUS in the bundle sheath (Fig. 1e–g). However, when nucleotides −980 to −394 upstream of the predicted translational start site were removed GUS was no longer detectable in bundle sheath cells (Fig. 1e, f). Consistent with this, MUG assays showed a statistically significantly reduction in activity when these nucleotides were absent (Fig. 1g). Thus, nucleotides spanning −980 to −394 of the *SiR* promoter are necessary for bundle sheath specific expression.

To test whether this region is sufficient for bundle sheath specific expression, we linked it to the minimal CaMV35S core promoter. Although weak GUS signal was detected in a few veinal cells, this was not the case for the bundle sheath (Fig. 1e–g). We conclude that sequence in two regions of the promoter (from −394 to +42 and from −980 to −394) interact to specify expression to the bundle sheath. To better understand this interaction, we next generated unbiased 5′ and 3′ deletions. This second deletion series further reinforced the notion that the *SiR* promoter contains a complex *cis*-regulatory landscape. For example, when nucleotides −980 to −829 were removed very weak GUS staining was observed and the MUG assay showed that activity was reduced by 73% (Fig. 2, Supplementary Fig. 5). We conclude that nucleotides −980 to −829 from the *SiR* promoter are necessary for tuning expression in the leaf. When nucleotides −829 to −700 were removed GUS appeared in mesophyll cells (Supplementary Fig. 5). Truncating nucleotides −613 to −529 diminished GUS accumulation (Supplementary Fig. 5). The 3′ deletion that removed nucleotides −251 to +42 also stopped accumulation of GUS in both bundle sheath and mesophyll cells (Fig. 2a–c, Supplementary Fig. 5). Notably, when the distal region required for bundle sheath expression (−980 to −829) was combined with nucleotides −251 to +42 this was sufficient for patterning to the bundle sheath (Fig. 2a, b).

Having identified a region in the *SiR* promoter that was necessary and sufficient for patterning to the bundle sheath, we next used phylogenetic shadowing and yeast one hybrid analysis to better understand the *cis*-elements and *trans*-factors responsible. Analysis of *cis*-elements in the *SiR* promoter that are highly conserved in grasses identified a short region located from nucleotides −588 to −539 that contained an *ETHYLENE INSENSITIVE3-LIKE 3* (*EIL3*) transcription factor binding site (Supplementary Fig. 6a, b). Whilst deletion of this motif had no detectable effect of patterning to the bundle sheath (Supplementary Fig. 6c), the level of expression was reduced (Supplementary Fig. 6d). We infer that the *EIL3* motif positively regulates activity of the *SiR* promoter but is not responsible for cell specificity. These data are consistent with the promoter truncation analysis that showed nucleotides −613 to −529 containing this motif were not required for bundle sheath specific expression, but instead function as a constitutive activator (Supplementary Fig. 5). When yeast one hybrid was

used to search for transcription factors capable of binding the *SiR* promoter, sixteen were identified (Supplementary Fig. 7a, b). For each, cognate binding sites were present. This included TCP21 and OsOBF1 that can bind to TCP motifs and Ocs/bZIP elements respectively. Consistent with the outcome of deleting the *EIL3* motif, three *EIL* transcription factors interacted with nucleotides −899 to −500 (Supplementary Fig. 7b, c). Examination of transcript abundance in mature leaves showed that most of these transcription factors were expressed in both bundle sheath and mesophyll cells (Supplementary Fig. 7d), implying that combinatorial interactions with cell-specific factors are likely required for bundle sheath-specific expression from the *SiR* promoter.

## The CRM contains four subregions that simultaneously activate in bundle sheath and repress in mesophyll cells

The truncation analysis above identified two short regions comprising nucleotides −980 to −829 and −251 to +42 that were necessary and sufficient for expression in the rice bundle sheath (Fig. 2, Supplementary Fig. 8). Sequence spanning nucleotides −251 to +42 includes both the annotated 5′ untranslated region but also likely contains core promoter elements (Supplementary Fig. 9a). Reanalysis of publicly available data identified two major transcription start sites at positions −91 (TSS1) and −41 (TSS2) (Supplementary Fig. 9a). Although no canonical TATA-box was evident in this region, a TATA-box variant was detected at position −130 (5′-ATTAAA-3′)[45] that could be responsible for transcription from TSS1. Upstream of TSS2 is a putative pyrimidine patch (Y-patch) that represents an alternate but common TC-rich core promoter motif in plant genomes)[45] (Supplementary Fig. 9a). Scanning sequence from −251 to −1 for core promoter elements also identified MTE (Motif Ten Element), BREu (TFIIB Recognition Element upstream), and DCE-S-I (Downstream Core Element S-I) motifs associated with eukaryotic core promoters (Supplementary Fig. 9b). We therefore assume the region upstream of TSS1 and TSS2 contains core promoter elements. When consecutive deletions to this sequence were made, statistically significant reductions in MUG activity were evident but there was no impact on accumulation of GUS in the bundle sheath (Supplementary Fig. 9c, d). Interestingly, when the Y-patch was retained but the TATA-box like motif removed, GUS was still detected in the bundle sheath (Supplementary Fig. 9c-f), but after deletion of the Y-patch GUS staining was no longer detectable in these cells (Supplementary Fig. 9c-f). Consistent with the Y-patch being important for bundle sheath expression, two core promoters with only a TATA-box linked to the distal CRM did not generate detectable expression in the bundle sheath, but those from genes with one or more Y-patches did (Fig. 3a, b). GUS activity was higher from the *PIP1;1* core promoter that contains three Y-patches (Fig. 3c). Overall, we conclude that the TATA-box like motif is not required for expression in the bundle sheath, but the Y-patch is necessary for this patterning and in combination with a distal CRM comprising nucleotides −980 to −829, it is sufficient for expression in this cell type.

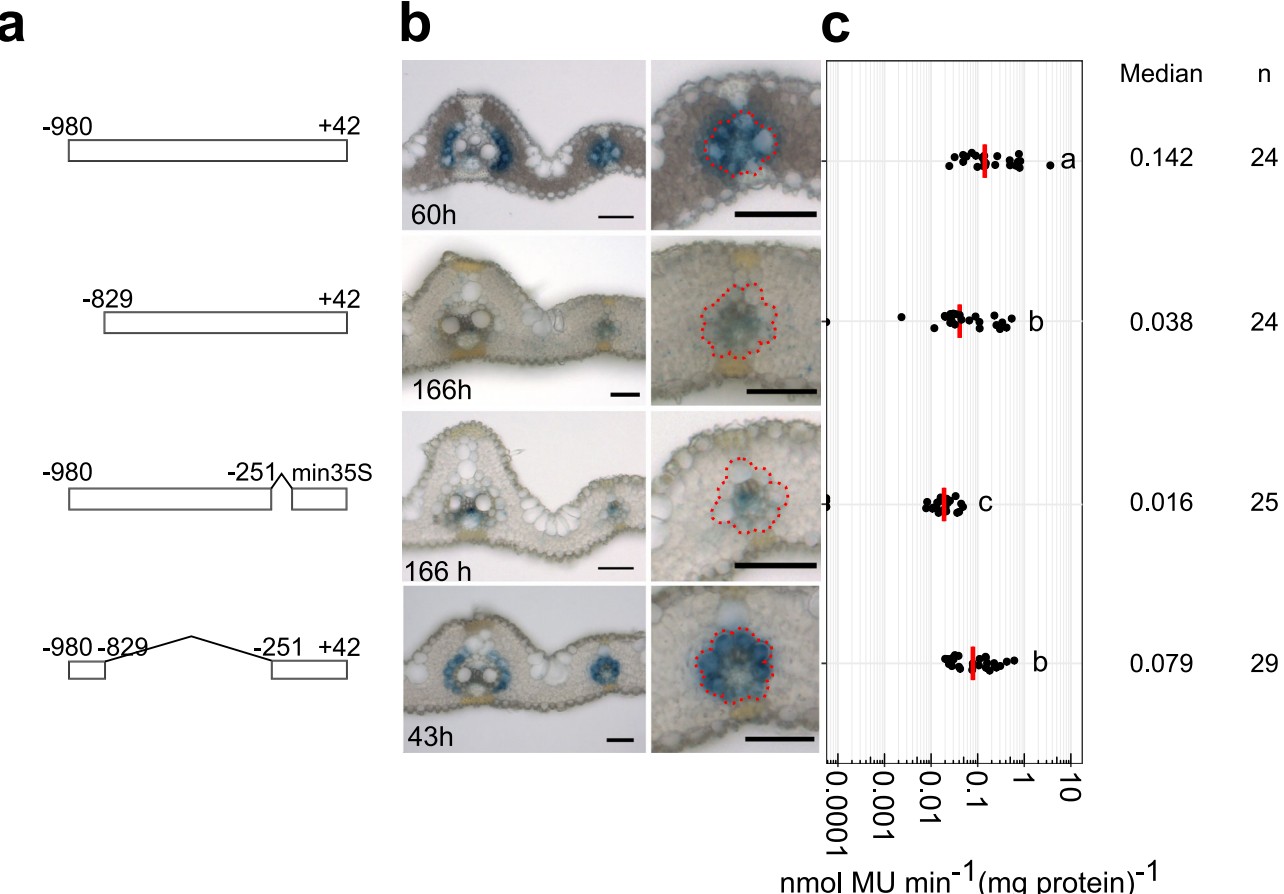

**Fig. 2 | A distal cis-regulatory module (CRM) and the core promoter that are necessary and sufficient for bundle sheath expression. a** Schematics showing deletions of nucleotides −980 to −829 and −251 to +42. **b** Representative image of leaf cross sections of transgenic lines after GUS staining. Zoomed-in images of lateral veins shown in right panels, the staining duration is displayed in the bottom-left corner, bundle sheath cells highlighted with dashed red line, scale bars = 50 μm. **c** Promoter activity determined by the fluorometric 4-methylumbelliferyl-β-D-

glucuronide (MUG) assay. Data subjected to pairwise two-sided Wilcoxon rank-sum test with Benjamini-Hochberg correction for multiple comparison. Lines with differences in activity that were statistically significant (adjusted $P < 0.05$) were labelled with different letters. The median represents the median GUS activity value and is shown as red line in the plot, n indicates the number of $T_0$ transgenic plants analysed. Source data are provided as a Source Data file.

We assessed the distal CRM for transcription factor binding sites. The FIMO algorithm identified motifs associated with WRKY, G2-like, MYB-related, MADS, DOF, IDD, ARR, and SNAC (Stress-responsive NAC) families. PlantPAN[46], which includes experimentally validated *cis*-elements, found an additional Dc3 Promoter Binding Factor (DPBF) binding site for group A bZIP transcription factors[47] (Fig. 3d). Seven consecutive deletions spanning this CRM region and hereafter termed subregions a-g were generated (Fig. 3d). Although veinal expression persisted when subregions a, b, and d were absent, deletion of subregions a, b, d and f resulted in loss of GUS from bundle sheath cells (Fig. 3e–g). MUG analysis showed that deletion of all four regions significantly reduced promoter activity (Fig. 3g). In contrast, deletions of nucleotides −938 to −923 (subregion c), −904 to −873 (subregion e), and −853 to −829 (subregion g) had no impact on the patterning (Supplementary Fig. 10). The subregions necessary for expression in the bundle sheath contained binding sites for WRKY, G2-like, MYB-related, DOF, IDD, SNAC, and bZIP (DPBF) transcription factors. To examine the significance of these regions in the context of full-length *SiR* promoter, consecutive deletions from subregion a to f were generated (Supplementary Fig. 11a). Deletion of subregion a, d or f, led to GUS accumulating primarily in mesophyll cells whereas removal of subregion b, c or e, caused GUS staining in both mesophyll cells and bundle sheath cells (Supplementary Fig. 11b). No significant changes in GUS activity were observed in these deletion lines (Supplementary

Fig. 11c). We conclude that the distal CRM generates expression in the bundle sheath due to four distinct sub-regions, and that by interacting with nucleotides −829 to −251, nucleotides between -980 to -853 also function as repressors of mesophyll expression.

## WRKY, G2-like, MYB-related, DOF, IDD and bZIP transcription factors activate the distal CRM

To gain deeper insight how the distal CRM operates, we employed transactivation assays, co-expression analysis and site-directed mutagenesis. The distal CRM contained WRKY, G2-like, MYB-related, DOF, IDD, SNAC, and bZIP (DPBF) motifs (Fig. 4a, Supplementary Fig. 12). DOF transcription factors have recently been shown to tune expression in bundle sheath cells[27] and so using an effector assay, we tested whether the other families interacted with the distal CRM. Using publicly available data, selection was based on three criteria: first, co-expression with *SiR*; second, preferential expression in bundle sheath cells from mature rice leaves; third, rice orthologs of Arabidopsis transcription factors predicted by FIMO to have the strongest binding to each motif (Supplementary Fig. 13). WRKY121, GLK2, MYBS1, IDD2/3/4/6/10, and bZIP3/4/9/10/11 transcription factors led to the strongest activation of expression from the bundle sheath CRM (Fig. 4b, Supplementary Fig. 14a–d), whereas the stress-responsive NAC transcription factors targeting a SNAC motif that overlaps a bZIP (DPBF) motif, activated less strongly than bZIP factors (Supplementary Fig. 14e). We

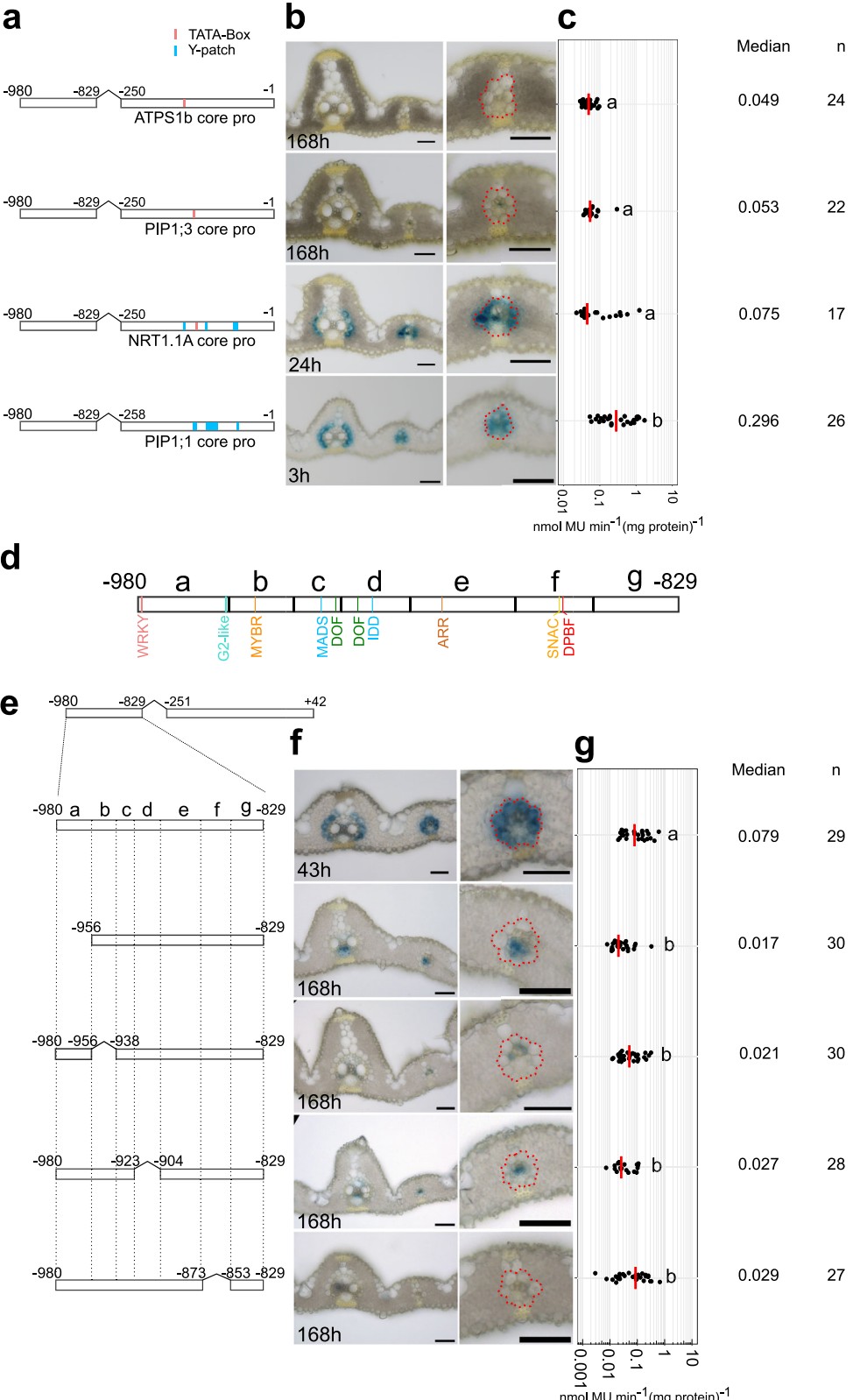

therefore conclude that the SNAC motif is not important for activity of the bundle sheath CRM. Effector assays using pairwise combinations of transcription factors showed synergistic activation from the distal CRM when GLK2 and IDD3,4,6,10 were co-expressed (Fig. 4c, Supplementary Fig. 14f).

Co-expression analysis derived from a cell-specific leaf developmental gradient dataset (Supplementary Data 5) revealed that

transcripts derived from *GLK2, MYBS1* and *IDD4,6,10* transcription factors that bind the G2-like, MYB-related and IDD motifs respectively were more abundant in mesophyll cells (Fig. 4d). However, transcripts for *bZIP9, IDD2* and *WRKY121* transcription factors strongly correlated with *SiR* transcript abundance and were preferentially expressed in bundle sheath cells (Fig. 4d). To test for sufficiency, we mis-expressed *bZIP9, IDD2, WRKY121* separately and in multiple combinations in

**Fig. 3 | The Y-patch and four distinct regions in the distal cis-regulatory module (CRM) are required for bundle sheath specific expression. a–c,** Nucleotides −980 and −829 from the *SiR* promoter pattern expression to the bundle sheath when linked with the *PIP1;1* and *NRT1.1 A* core promoters containing Y-patches. **a** Schematics showing Y-patch and TATA-box in core promoters of *ATPSb, PIP1;3, NRT1.1 A* and *PIP1;1* which have been used to initiate the transcription of nucleotides −980 and −829. **b** Representative cross sections of transgenic rice leaves after GUS staining, zoomed-in image of lateral veins shown in the right panel, bundle sheath cells highlighted with red dashed lines, the staining duration is displayed in the bottom-left corner, scale bars = 50 μm. **c** Promoter activity determined by the fluorometric 4-methylumbelliferyl-β-D-glucuronide (MUG) assay. **d** Schematics showing transcription factor binding sites between nucleotides −980 and −829.

**e** Schematics showing consecutive deletions between nucleotides −980 and −829 fused to the GUS reporter. **f** Representative images of cross sections from transgenic lines after GUS staining, zoomed-in images of lateral veins shown in right panels, the staining duration is displayed in the bottom-left corner, bundle sheath cells highlighted with red dashed lines, scale bars = 50 μm. **g** Promoter activity determined by the fluorometric 4-methylumbelliferyl-β-D-glucuronide (MUG) assay. In (**c, g**) data were subjected to pairwise two-sided Wilcoxon rank-sum test with Benjamini-Hochberg correction. Lines with differences in activity that were statistically significant (adjusted $P < 0.05$) were labelled with different letters. Median catalytic rate of GUS indicated with red lines, *n* indicates total number of $T_0$ transgenic plants assessed. Source data are provided as a Source Data file.

mesophyll cells. Mesophyll expression of *bZIP9* or *WRKY121* induced GUS expression from the bundle sheath CRM in some mesophyll cells, with the effect from *WRKY121* being greater (Fig. 4e, Supplementary Fig. 15b, f). Mesophyll expression of *IDD2* suppressed activity (Fig. 4e, Supplementary Fig. 15d), but when combined with *bZIP9* significant expression in the mesophyll was apparent (Fig. 4e, Supplementary Fig. 15h). Strikingly, expression of *bZIP9, IDD2* and *WRKY121* in the mesophyll strongly activated expression in this cell type (Fig. 4e, Supplementary Fig. 15j). When each motif was mutated, with the exception of the WRKY, CRM activity in the bundle sheath was diminished (Fig. 5a–c).

In order to test whether the distal CRM is sufficient to pattern expression to rice bundle sheath cells, we concatenated a sequence containing the WRKY, G2-like, MYB-related, DOF, IDD, and bZIP sites and fused them to the core promoter of *SiR* (Fig. 5d). GUS staining was evident in the bundle sheath (Fig. 5e). Fusion to the *PIP1;1* core promoter maintained bundle sheath expression and resulted in an ~5 fold increase in activity (Fig. 5d–f). Oligomerisation of the CRM by repeating it three or five times increased bundle sheath specific expression 25 or 58-fold respectively, when fused to *SiR* core promoter, and this effect was amplified 94 and 224-fold when fused with the *PIP1;1* core (Fig. 5d–f). When an oligomerised version of the CRM was linked to the *SiR* core promoter and placed in *A. thaliana*, it generated strong expression in bundle sheath cells (Fig. 5g, Supplementary Fig. 16).

Collectively, our data reveal an ensemble of transcription factors belonging to the WRKY, G2-like, MYB-related, DOF, IDD, and bZIP (DPBF) families act to decode distinct *cis*-elements in a distal CRM of the *SiR* promoter, and that this transcription factor team represents an ancient and highly conserved mechanism allowing bundle sheath-specific gene expression in both monocotyledons and dicotyledons.

## Discussion
### Expression of multiple genes in the rice bundle sheath is not associated with close upstream CRM
Gene expression is determined by interactions between elements in the core promoter, allowing basal levels of transcription[48,49] with more distal *cis*-regulatory modules[50–52]. Such *cis*-regulatory modules include enhancers and silencers that act as hubs receiving input from multiple transcription factors and so allow gene expression to respond spatially and temporally to both internal and external stimuli[53,54]. After testing 25 promoters, we discovered that the majority were not capable of driving expression in the rice bundle sheath, and this included ten that generated no detectable activity of GUS in leaves. In all cases, we had cloned sequence between −3191 and −960 nucleotides upstream of the predicted translational start site and so these data demonstrate that the core promoter and any CRM in these regions are not sufficient to direct expression to rice bundle sheath cells. Combined with the paucity of previously reported promoters active in this cell type[55,56], these data argue either for long-range upstream enhancers[57–61] or other regulatory mechanisms being important to specify expression in the bundle sheath. Possibilities include transcription factor binding sites in introns that impact on transcription start site and strongly enhance

gene expression[62,63], or in exons where because such sequences specify amino acid sequence as well as binding of *trans*-factors, they have been termed duons[64]. Functional analysis showed that duons can pattern expression to the bundle sheath of the $C_4$ plant *Gynandropsis gynandra*[65], and it is notable that a genome-wide analysis of transcription factor binding sites in grasses revealed genes preferentially expressed in bundle sheath cells tended to contain transcription factor binding sites in coding sequence[66]. It therefore appears possible that gene expression in the bundle sheath is commonly encoded by non-canonical architecture perhaps based on duons rather than more traditional cis-regulatory elements upstream of the core promoter.

Despite the above, we discovered four promoters capable of driving expression in the rice bundle sheath, and each was associated with a gene important in sulphur metabolism. For example, *ATPS1b, SiR,* and *Fd* all participate in the first two steps of sulphate reductive assimilation[67–69], while *HAC1;1* encodes an arsenate reductase important in the detoxification of arsenate using glutathione that is a product of sulphur assimilation[70]. Collectively, these data support the notion that various mechanisms underpin bundle sheath specific expression in rice, but that for some genes involved in sulphur metabolism upstream regulatory regions are important.

### Two distinct genetic networks governing expression in bundle sheath cells
The only other promoter for which both *cis*-elements and *trans*-factors that are necessary and sufficient to pattern bundle sheath expression have been reported is from the dicotyledonous model *A. thaliana*. In that study, a bipartite MYC-MYB module upstream of the *MYB76* gene is responsible for this output[71]. MYB76 forms part of a network governing glucosinolate biosynthesis in *A. thaliana*, and so it is notable that the gene regulatory network we report in rice is also associated with sulphur metabolism. However, rather than the bipartite transcription factor module that activates *MYB76* in the *A. thaliana* bundle sheath, in rice, we report a more complex regulatory landscape where a bundle sheath CRM is embedded in constitutive activators and an overlapping mesophyll silencer (Fig. 6a–c). This complexity may be associated with the fact that SiR is considered to catalyse the rate-limiting step for sulphur assimilation[68]. The CRM controlling bundle sheath *SiR* expression in rice comprises four distinct regions recognised by transcription factors belonging to the WRKY, G2-like, MYB-related, DOF, IDD and bZIP families (Fig. 6d). It is of course possible that additional motifs in the CRM modulate the level of expression in bundle sheath cells. As loss of the G2-like, MYB-related, DOF, IDD and bZIP motifs all reduced expression in the bundle sheath, this implies they act cooperatively—a notion further supported by the fact that GLK2 and IDD3,4,6,10 synergistically activated promoter output in a transient assay. This cooperation between transcription factors may explain the lack of overlap between these transcription factors and ones identified by the yeast one-hybrid assay, as the latter is typically not well suited to identify transcription factor complex-DNA interactions[72]. To examine whether other bundle sheath-expressed genes may be regulated by the CRM and Y-patch system we report, we

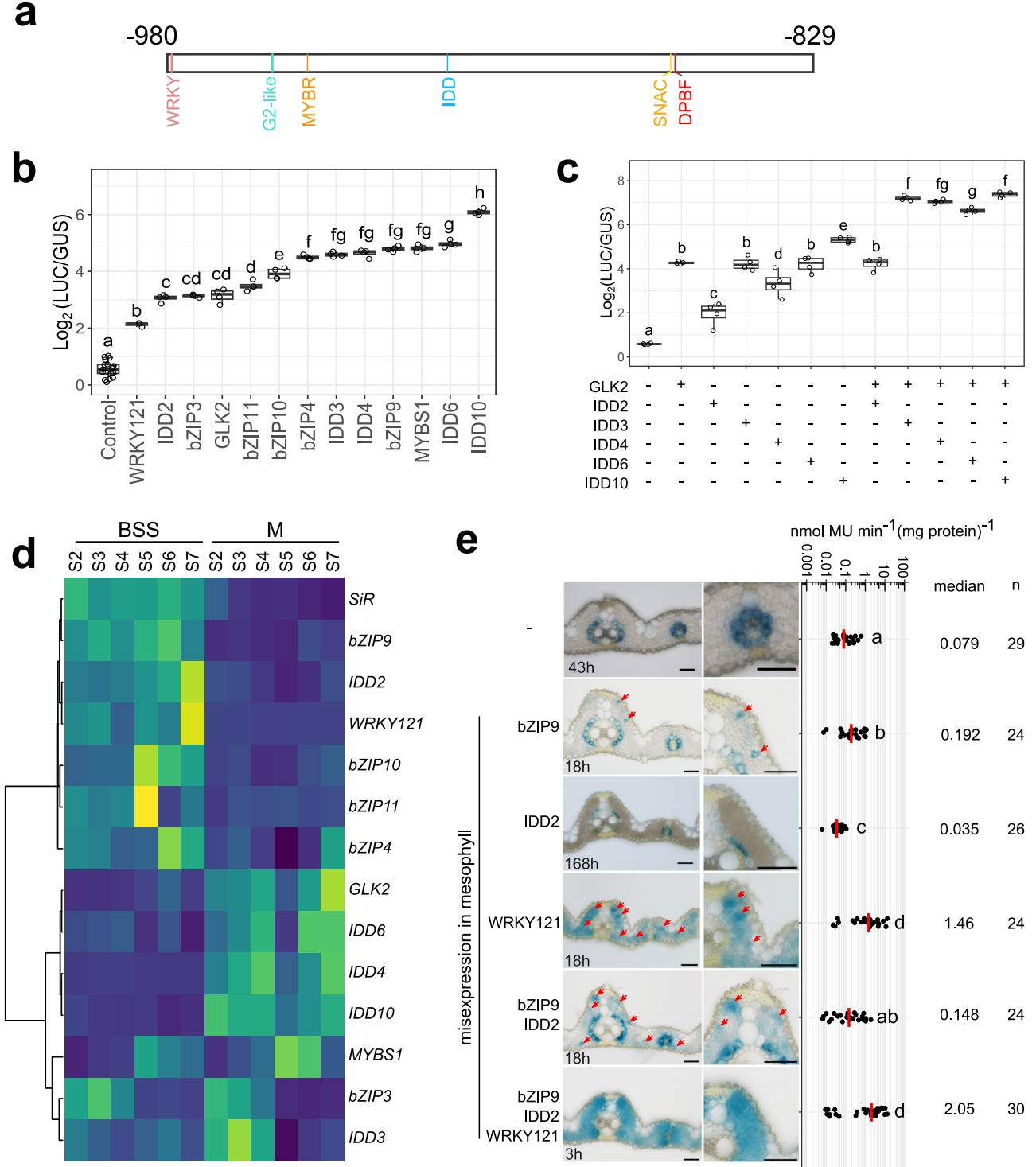

**Fig. 4 | WRKY, G2-like, MYB-related, IDD and bZIP transcription factors interact and activate with the distal cis-regulatory module (CRM). a** Schematics showing transcription factor binding sites between nucleotides −980 and −829, which are likely required for bundle sheath-specific expression. **b** Effector assays showing that each transcription factor activates expression from the distal CRM. **c** Effector assays showing synergistic activation from the distal CRM when GLK2 and IDD3, 4, 6, 10 were co-expressed. In (**b**) and (**c**), LUC/GUS ratio from four biological replicates were $\log_2$ transformed and analysed using pairwise t-tests with Benjamini-Hochberg correction for multiple comparisons. Statistically significant differences (adjusted $P < 0.05$) are indicated by different letters. Box plots show the 25th, 50th, and 75th percentiles; whiskers extend to the most extreme values within 1.5× the interquartile range. The assay was independently repeated three times with similar results. Source data are provided as a Source Data file. **d** Transcript abundance of transcription factors in bundle sheath strands (BSS) and mesophyll (M) cells during maturation. Leaf developmental stage S2 to S7 represent base of the 4th leaf at the 6th, 8th, 9th, 10th, 13th and 17th day after sowing. **e** Representative images of transgenic lines mis-expressing WRKY121, IDD2 and bZIP9 in mesophyll cells, staining duration is displayed in the bottom-left corner, zoom-in of mesophyll shown in right panel, red arrows indicate GUS expressing mesophyll cells. The bundle sheath CRM activity is determined by the fluorometric 4-methylumbelli-feryl-β-D-glucuronide (MUG) assay, data subjected to pairwise two-sided Wilcoxon rank-sum test with Benjamini-Hochberg correction. Lines showing statistically significant differences in activity (adjusted $P < 0.05$) were labelled with different letters. The median represents the median GUS activity value and is shown as red lines in the plot, n indicates the number of $T_0$ transgenic plants analysed. Source data are provided as a Source Data file.

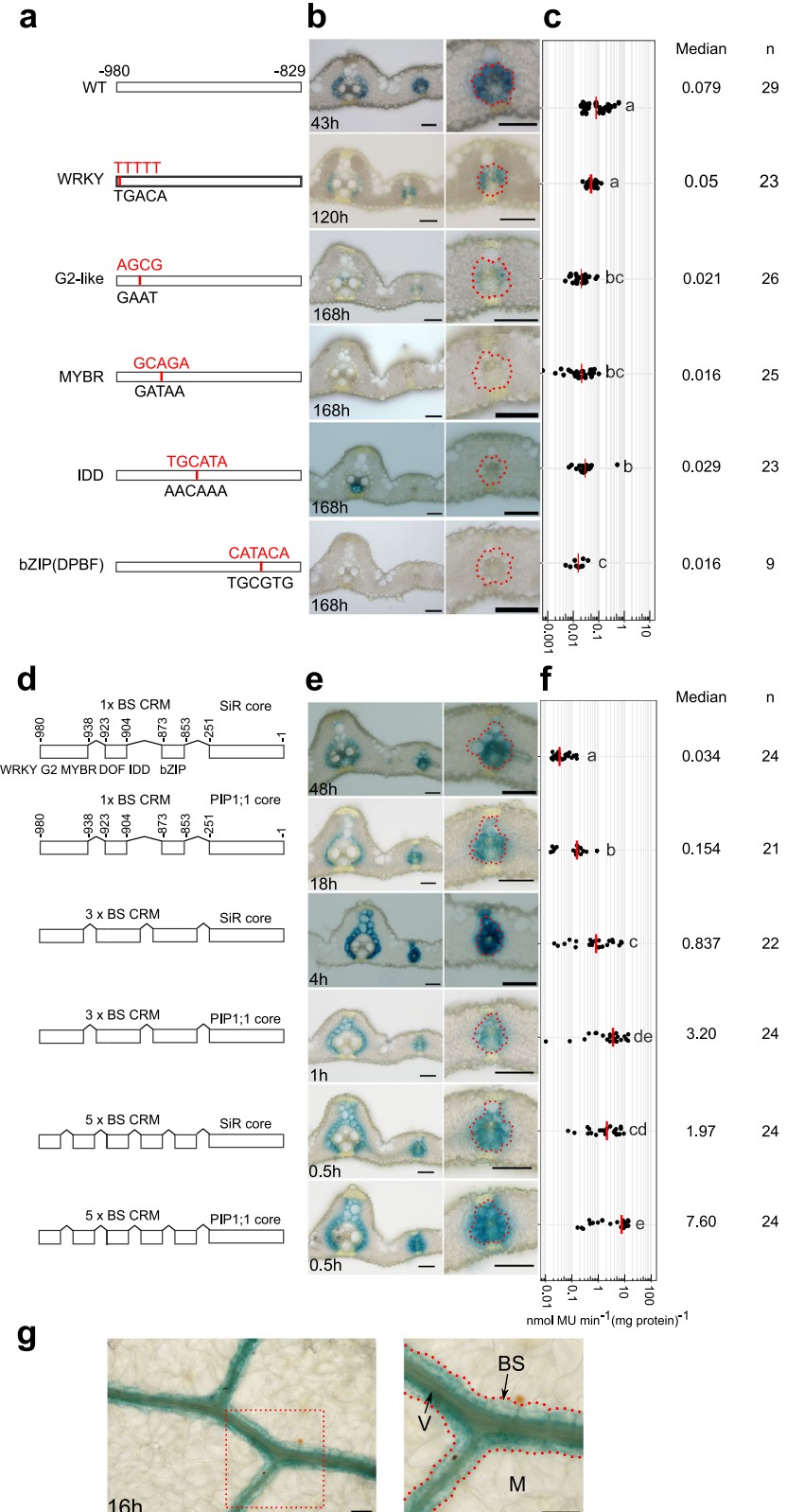

assessed their prevalence in open chromatin across the genome. This demonstrated over-representation of these sequences in bundle sheath-expressed genes (Supplementary Fig. 18). Of the 283 genes containing both Y-patch and *cis*-elements found in the CRM of *SiR*, 117 were expressed in the bundle sheath, including 21 preferentially expressed in this cell type. Interestingly, this included *a Ferredoxin* annotated as providing reducing equivalents to *SiR*[67], *a CRT-like*

*transporter 1* (*CLT1*) allowing glutathione transport[73], a sulfurtransferase *STR22*[74], and two transcription factors *EIL2* and *EIL3* involved in ethylene signalling[75], but also considered master regulators of sulphur assimilation[76,77] (Supplementary Data 10).

The distal CRM in the *SiR* promoter operates in conjunction with the core promoter that contains two transcription start sites, one with an upstream TATA-box and the other a TC-rich element known as a

**Fig. 5 | Oligomerisation of bundle sheath cis-regulatory module (CRM) increases bundle sheath expression. a, d** Schematics showing site-directed mutagenesis of WRKY, G2-like, MYBR, IDD and bZIP motifs, mutated nucleotides highlighted in red (**a**), and constructs to test the impact of oligomerization of the CRM (**d**). **b, e** Representative images of cross sections from transgenic lines after GUS staining, zoomed-in images of lateral veins shown in right panel, the staining duration is displayed in the bottom-left corner, bundle sheath cells highlighted with red dashed lines, scale bars = 50 μm. **c, f** Promoter activity determined by the fluorometric 4-methylumbelliferyl-β-D-glucuronide (MUG) assay. Data subjected to

pairwise two-sided Wilcoxon rank-sum test with Benjamini-Hochberg correction. Lines with differences in activity that were statistically significant (adjusted $P < 0.05$) were labelled with different letters. Median catalytic rate of GUS indicated with red lines, n indicates total number of $T_0$ transgenic plants assessed. Source data are provided as a Source Data file. **g** Paradermal view of Arabidopsis leaf expressing GUS under the control of 3x BS CRM combined with *OsSiR* core promoter, the staining duration is displayed in the bottom-left corner. M indicate mesophyll, BS for bundle sheath, and V for vein. Zoomed in images for the red box shown on right, bundle sheath highlighted with red dash lines.

pyrimidine (Y) patch (Supplementary Fig. 9a). The TATA-box is found in metazoans and plants and allows recognition by the pre-initiation complex[78], but in plants, computational analysis showed that many promoters lack a TATA-box and instead contain a Y-patch[79–81]. These genes tend to be relatively steadily expressed and associated with protein metabolism[81], and presence of a Y-patch can increase core promoter strength[82]. For *SiR*, whilst the TATA-box is not required, the Y-patch is needed for expression in the bundle sheath. Notably, both synthetically generated (Supplementary Fig. 17) and endogenous core promoters modified to contain more Y-patches tended to drive stronger expression, indicating that in plants cell, specific gene expression could be tuned by selecting different core promoters.

The distal CRM (nucleotides -980 to -829) also represses mesophyll expression if nucleotides −829 to −700 are present. This suggests these two regions interact to suppress transcription in the mesophyll (Fig. 6a–c). Thus, a mesophyll silencer overlaps with the bundle sheath CRM (Fig. 6c). Of the six families of transcription factors binding the bundle sheath CRM, WRKY121, IDD2, and bZIP9 appear critical in controlling bundle sheath-specific expression because misexpression in the mesophyll leads to output in this cell type (Fig. 6d). It is possible that transcription factors more highly expressed in the mesophyll such as GLK2, MYBS1 and IDD4/6/10 interact with additional factors binding to the −829 to −700 region and that this contributes to mesophyll-specific suppression (Fig. 6e). In addition to controlling cell specificity, this complexity likely also facilitates the tuning of expression to environmental conditions. For instance, the *EIL* motif (position −588 to −539) is recognised by *ETHYLENE-INSENSITIVE LIKE* transcription factors that respond to sulphur deficiency[76,77]. As transcripts encoding *EIL* accumulate in both bundle sheath and mesophyll cells in response to sulphate deficiency, it seems likely that transcription factors repressing expression in the mesophyll respond in a dynamic manner. In addition to *EIL*, the yeast one hybrid analysis identified seven other families of transcription factor families that can bind the *SiR* promoter. Many play documented roles during abiotic or biotic stress, with for example OBF1, ERF3, NAP and FLP acting during low-temperature or drought responses[83–87], while TCP21, EREBP1, ERF3, ERF72, and ERF83 are involved in both abiotic and biotic stress[88–91]. Consistent with previous in silico analysis[92] the presence of multiple AP2/ERF and EIL transcription factors binding sites suggests that *SiR* is likely subject to control from ethylene signalling[93] and also of transcription factors that respond to abscisic acid (ABA) and jasmonic acid (JA)[86,90,94–96]. Together this implies that multiple phytohormone signalling pathways converge on the *SiR* promoter. These data are similar to those reported for the *SHORTROOT* promoter in *A. thaliana* roots where a complex network of activating and repressing *trans*-factors also tunes expression[97]. It is also notable that the architecture we report for the bundle sheath CRM of *SiR* appears of similar complexity to the collective of five transcription factors used to specify cardiac mesoderm in *Drosophila melanogaster* and vertebrates[3]. For the five transcription factors that bind the cardiac mesoderm enhancer, the order and positioning of motifs (motif grammar) is flexible. However, this is not always the case, with for example output from the human *interferon-beta* (*INF-β*) enhancer demanding a conserved grammar[98,99]. Further work will be needed to determine if the bundle sheath CRM reported here is more similar to one of these models, or indeed, as reported for

the Drosophila *eve stripe 2* enhancer, operates as a billboard in different tissues to determine patterning of expression[100].

## Using the *SiR* promoter to engineer the rice bundle sheath

In addition to bundle sheath cells being important for sulphur assimilation[28–30], they have also been implicated in nitrate assimilation, the control of leaf hydraulic conductance and solute transport[30] and the systemic response to high light[101]. Moreover, in one of the most striking examples of a cell type being repurposed for a new function, bundle sheath cells have repeatedly been rewired to allow the evolution of $C_4$ photosynthesis[31]. To engineer these diverse processes, specific and tuneable promoters for this cell are required. However, identification of sequence capable of driving specific expression to bundle sheath strands has previously been limited to *A. thaliana* and $C_4$ species. For example, the *SCARECROW*[402], *SCL23*[102], *SULT2;2*[103] and *MYB76* promoters[71] are derived from *A. thaliana*, whilst the *Glycine Decarboxylase P-protein (GLDP)* promoter is from the $C_4$ dicotyledon *Flaveria trinervia*[104,105]. In rice, only the $C_4$ *Zoysia japonica PCK* and the $C_4$ *Flaveria trinervia GLDP* promoters are known to pattern expression to the bundle sheath[55,56]. Both are capable of conditioning expression in this cell type, but are weak, turn on late during leaf development and the molecular basis underpinning their ability to restrict expression to the bundle sheath has not been defined. It has therefore not been possible to rationally design or tune expression to this important cell type in rice. The architecture of the *SiR* promoter we report here now provides an opportunity to engineer the bundle sheath.

In summary, from analysis of the -2600 nucleotide *SiR* promoter we identify an CRM comprising 81 nucleotides that, with the Y-patch is sufficient to drive expression to bundle sheath cells. Moreover, we show that output from the sequence can be tuned via two approaches. First, oligomerising the distal CRM can drastically increase expression. Second, combining it with different core promoters achieved the same output, and correlated with copy number of the Y-patch present in natural and synthetic promoters. Our identification of a minimal promoter that drives expression in bundle sheath cells of rice now provides a tool to allow this important cell type to be manipulated. Cell-specific manipulation of gene expression has many perceived advantages. For example, when constitutive promoters have been used to drive gene expression gene silencing and reduction of plant fitness due to metabolic penalties[106,107]. In contrast, tissue-specific promoters allow targeted gene expression either spatially or at particular developmental stages and so allow increased precision in trait engineering[108]. The *SiR* promoter and the bundle sheath *cis*-regulatory module that we identify thus provide insights into mechanisms governing cell specific expression in plants, and may also contribute to our ability to engineer and improve cereal crops.

## Methods

### Plant material and growth conditions

Kitaake (*O. sativa* ssp. *japonica*) was transformed using *Agrobacterium tumefaciens* as described previously[109] with the following modifications. Mature seeds were sterilised with 2.5% (v/v) sodium hypochlorite for 15 mins, and calli were induced on NB medium with 2 mg/L 2,4-D at 30 °C in darkness for 3-4 weeks. Actively growing calli were then co-incubated with *A. tumefaciens* strain *LBA4404* in darkness at 25 °C for 3 days, selected on NB medium supplied with 35 mg/L hygromycin B

for 4 weeks, and those that proliferated, placed on NB medium with 10 mg/L hygromycin B for 4 weeks at 28 °C under continuous light. Plants resistant to hygromycin were planted in 1:1 mixture of topsoil

and sand and placed in a greenhouse at the Botanic Garden, University of Cambridge under natural light conditions but supplemented with a minimum light intensity of 390 μmol m⁻² s⁻¹, a humidity of 60%,

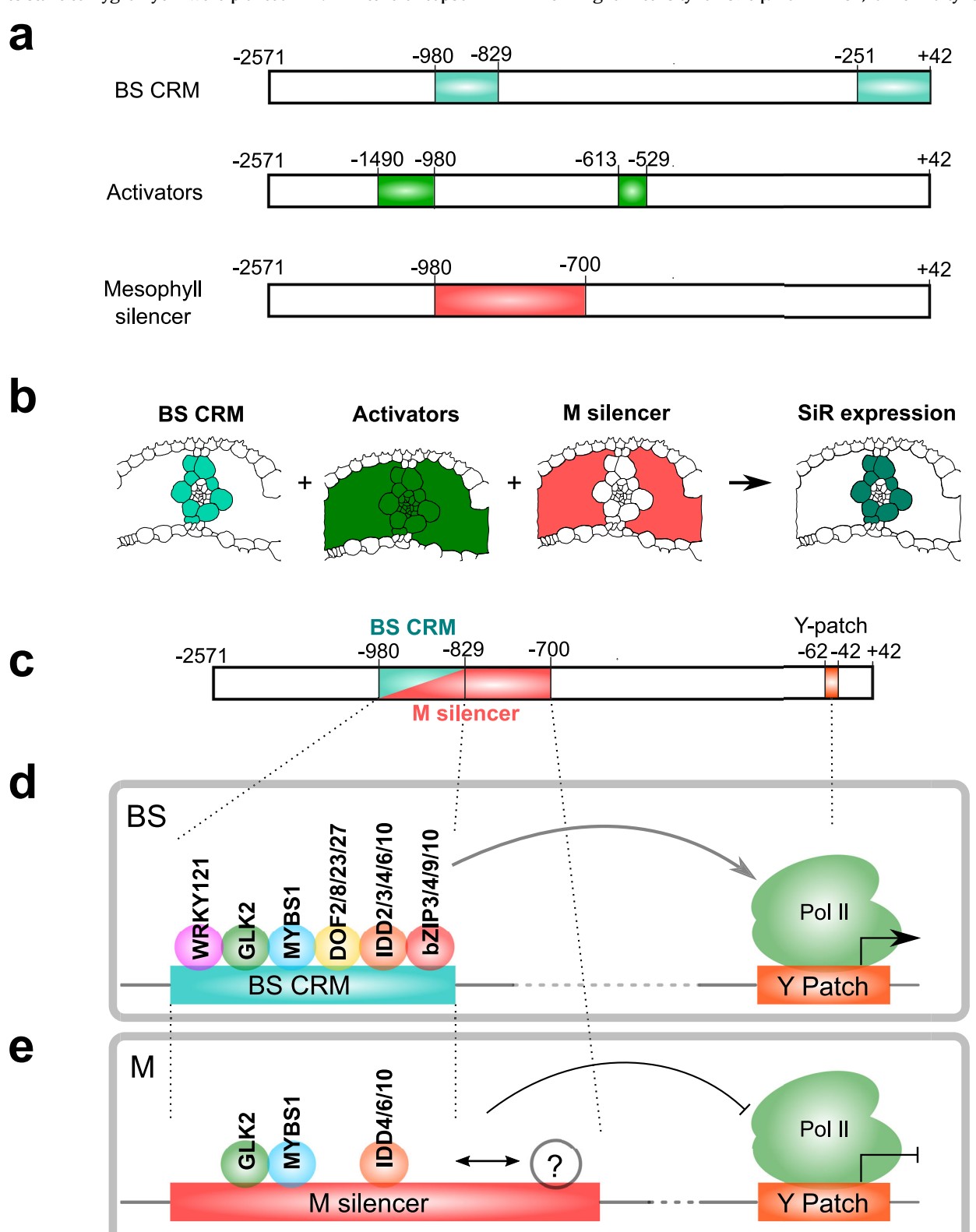

**Fig. 6 | Model of mechanism underpinning bundle sheath expression from *SiR* promoter. a** Schematic with location of the bundle sheath (BS) cis-regulatory module (CRM), constitutive activators and mesophyll silencer. **b** Bundle sheath expression is a result of the CRM, constitutive activators and mesophyll silencer acting in concert. Schematic indicating how the CRM operates within a broader *cis*-regulatory landscape. **c**–**e**, Model depicting transcription factors and cognate *cis*-elements responsible for bundle sheath expression (**d**) and mesophyll repression (**e**).

temperatures of 28 °C and 20 °C during the day and night respectively, and a photoperiod of 12 h light, 12 h dark. Subsequent generations were grown in a growth cabinet in 12 h light/12 h dark, at 28 °C, a relative humidity of 65%, and a photon flux density of 400 μmol m$^{-2}$ s$^{-1}$.

## Laser capture microdissection RNA-seq and data analysis

For the first Laser Capture Microdissection (LCM) RNA-seq analysis, the middle 1-cm of the fourth fully expanded IR64 (*Oryza sativa* ssp. *indica*) leaves were sampled 4 h after dawn. Leaf tissue was fixed with acetone and embedded into Steedman's wax[39]. Paradermal sections of 7μm were prepared with a microtome and mounted on PEN membrane slides (Applied Biosystems, LCM0522). Bundle sheath strands (bundle sheath as well as phloem and xylem cells) and mesophyll cells were isolated using CapSure Macro Caps (Applied Biosystems, LCM0211). To prepare cDNA libraries, 20-40 ng RNA was depleted for ribosomal RNA using Ribozero kit (Illumina), and then the 1$^{st}$ strand cDNA was generated and amplified using Ovation v2 RNA-seq system (Nugen) according to the user manual. Single stranded cDNA was digested using S1 Nuclease (Promega, M5761) then 100 ng of amplified cDNA sheared using a Covaris E220 focused ultra-sonicator. cDNA libraries were prepared using Truseq nano DNA library preparation kit (Illumina). Paired-ended 75-bp sequencing were performed using Nextseq 550 sequencer. Poly A/T/N and low-quality reads were removed using AfterQC and Trimmomatic, and gene expression in transcripts per million (TPM) quantified using Salmon v1.1.0[110]. Differential gene expression analysis was carried out using DESeq2[111], with differentially expressed genes being defined with an adjusted *P* < 0.05 and log2 fold change > 1.

For subsequent analysis, the middle 1-cm of the fourth fully expanded leaves from Kitaake (*Oryza sativa* ssp. *japonica*) were fixed and embedded. Bundle sheath, mesophyll, and veinal cells were separated, library preparation and RNA-seq data analysis were conducted according to Hua et al., 2021. To identify bundle sheath specific promoters, candidate genes from bundle sheath and vein or bundle sheath alone clusters were filtered based on pairwise comparisons among the three tissue types using DESeq2 and EdgeR[112] (*Log2FC(BS/M) > 2 & adjusted P (BS/M) < 0.01 & FDR (BS/M) < 0.01 & Log2FC(BS/V) > −0.5*) to identify 210 genes specifically expressed in bundle sheath and vein or bundle sheath alone (Supplementary Data 4).

A leaf developmental gradient was also used for LCM RNA-seq. Here, Kitaake leaf tissue was also fixed and embedded according to Hua et al., 2021[30]. Material from the shoot apical meristem and leaf four primordia (referred to as Stage 1) were isolated from 3-day-old seedlings. Subsequent samples were taken from the leaf blade 5-mm above the leaf ligule at day 6 (Stage 2), 8 (Stage 3), 9 (Stage 4), 10 (Stage 5), 13 (Stage 6) and 17 (Stage 7) after sowing, with the midrib removed before embedding into Steedman's wax. Prior to LCM, sections of Stage 2, 3, and 4 were treated with 1% iodine (w/v) in acetone for 1 min. 1.4–3 ng of RNA from each cell type (three to four biological replicates for each stage) were reverse transcribed, and the resultant cDNA amplified using the SMART-Seq v4 Ultra Low Input RNA Kit (Clontech) according to the manufacturer's instructions. 11 PCR cycles were used for all samples. 1 ng of amplified cDNA was used as input for library preparation using the Nextera XT DNA Library Preparation Kit (Illumina). Libraries were then sequenced using Illumina's NovaSeq 6000 sequencing platform to generate 31–50 million paired-ended 150-bp reads for each sample. Transcript abundance was determined after reads were quantified using Salmon v1.5.1[110] against *Oryza sativa* v7.0 transcripts from Phytozome V13[113] with selective alignment ("--validateMappings") enabled. Gene-level abundance (transcripts per million, TPM) was summarised using tximport 1.18.0[114], and genes with TPM > 1 in at least three samples of at least one developmental stage were defined as expressed genes. Co-expression network analysis was performed using Log$_2$ transformed TPM and Weighted Correlation Network Analysis (WGCNA (v.1.63))[115] with soft threshold of 12, minimal module size of 100 genes, MEDissThres cut-off of 0.1.

## Construct preparation

To test promoters for bundle sheath specific expression, a minimal of 1.5-kb upstream sequences from translational start site or entire intergenic region if it is shorter than 1.5-kb were amplified from genomic DNA with primers listed in Supplementary Data 7; in some cases, this was extended both upstream and downstream of the TSS to capture regulatory elements identified through DNase hypersensitive sites (DHS) at the Plant DHSs Database (https://plantdhs.org/)[38]. Gel-purified PCR products were cloned into a Gateway pENTR vector using directional D-TOPO Cloning Kit (Invitrogen, K240020), and the promoters recombined into the *pGWB3* expression vector and fused with the *GUS* gene using LR reaction. The resultant vector was transformed into *A. tumefaciens* strain *LBA4404* and then Kitaake.

To engineer the *SiRpro* such that it is compatible with the Golden gate system, four *Bsa*I or *Bpi*I restriction enzyme recognition sites at −214, −298, −1468, and −2309 nucleotide relative to translational start site were mutated from T to A as described[116] using PCR primers listed in Supplementary Data 8, PCR fragments were assembled into the *pAGM9121* vector[117] using Golden Gate reactions, and used to drive *kzGUS* (intronless *GUS*)[55] and *H2B-mTurquoise2* reporter genes using *Tnos* as a terminator. A 5' prime deletion series was generated using the domesticated *SiRpro* as template and prepared as level 0 PU modules. The 3' prime deletion series was prepared as level 0 P modules with the minimal *CaMV35S* promoter as the U module and linked with *kzGUS* and terminated with *Tnos*. To test *SiRpro* with the dTALE/STAP system[41], the 42-bp coding region was excluded, and the 2571-bp resultant fragment placed into a level 0 PU module EC14330 and used to drive *dTALE*. Two reporters were used. For the GUS reporter *kzGUS* was linked with *STAP62* and terminated with *Tnos*. In the fluorescent reporter construct, a chloroplast targeting peptide fused to *mTurquoise 2* was linked with *STAP4* and terminated with *Tact2*. In both constructs, *OsAct1pro* driving *HYG* (the hygromycin-resistant gene) was terminated with *Tnos* and used as the selection marker during rice transformation.

Consecutive deletions and site-directed mutagenesis within nucleotides −980 to −829 were cloned as level 0 PU modules. Core promoter sequences from *SiR, PIP1;1, PIP1;3, NRT1.1 A,* and *ATPS1b* were amplified using primers listed in Supplementary Data 8 and cloned as level 0 U modules. For oligomerisation, the bundle sheath CRMs was cloned as level 0 P modules, and then assembled into level 1 modules with respective U modules as well as *kzGUS* and the *Tnos* terminator. To mis-express *bZIP9, IDD2,* and *WRKY121* individually or in combination in mesophyll cells, the *ZmPEPC* promoter[118,119] was used to drive *dTALE*, expression of *bZIP9, IDD2,* and *WRKY121* controlled by different STAPs: *bZIP9* under STAP62, *IDD2* under STAP45, and *WRKY121* under STAP56. When *IDD2* was expressed alone its was driven by the *ZmUBI* promoter. A reporter construct containing nucleotides −980 to −829 and the endogenous core promoter (-250 to +42) driving kzGUS was used to assess the CRM activity in the same level 2 constructs used to mis-express each transcription factor.

## Motif analysis

The Find Individual Motif Occurrences (FIMO) tool[43] from the Multiple Em for Motif Elucidation (MEME) suite v.5.4.0[120] was used to search for individual motifs within promoter sequences using default parameters with "--thresh" of "1e-3". Position weight matrix of 656 non-redundant plant motifs and 13 RNA polymerase II (POLII) core promoter motifs were obtained from JASPAR[44], the *DPBF* binding sites and the Y-patch was included as previously described[47,121]. To cluster the transcription factor binding motifs, the RSAT matrix-clustering tool[122] was run on all 656 non-redundant plant motifs using the default parameters. This yielded 51 motif clusters, that were classified based on transcription factor families (Supplementary Data 2).

To assess the co-occurrence of the Y-patch motif and the sextet of *cis*-elements present in the distal CRM of *SiR* across all genes in the rice

genome, we first examined the presence of the Y-patch motif in the core promoter. Specifically, we extracted DNA regions (5′ to 3′) spanning from 200 bp upstream of the transcriptional start site (TSS) to the translational start site and scanned this for the Y-patch motif using FIMO. The Y-patch motif was defined based on a p-value threshold of <0.0004 and was required to be in the same orientation as the gene. To investigate the co-occurrence of the sextet of motifs, we extracted accessible chromatin as defined by *DNase* I Hypersensitive Sites[38] within 2000 bp upstream of the TSS to the end of the 3′ UTR. Motif scanning was performed using FIMO with default parameters and a significance threshold ("--thresh") of 1e-3. The presence of the WRKY and G2-like motif families was determined using a p-value cutoff of <1e-3, whereas MYBR_B, DOF, IDD, and DPBF motif families were identified using a p-value cutoff of <1e-4 within a 300-bp window. Finally, 283 genes containing both the Y-patch motif in the core promoter and the distal CRM across the promoter and gene body were identified (Supplementary Data 10).

## Analysis of GUS and fluorescent reporters

In all cases, to account for position effects associated with transformation via *A. tumefaciens*, multiple $T_0$ lines were assessed for each construct. GUS staining was performed as described previously[123] with the following minor modifications. Leaf tissue was fixed in 90% (v/v) acetone overnight at 4 °C, after washing with 100 mM phosphate buffer (pH 7.0), leaf samples were transferred into 1 mg/ml 5-bromo-4-chloro-3-indolyl glucuronide (X-GlcA, Melford, B72200) GUS staining solution, subjected to 2 mins vacuum infiltration 5 times, and then incubated at 37 °C for between 0.5 and 168 h. Chlorophyll was cleared further using 90% (v/v) ethanol overnight at room temperature. Cross sections were prepared manually using a razor blade and images were taken using an Olympus BX41 light microscope. Quantification of GUS activity was performed using a fluorometric MUG assay[123]. ~200 mg mature leaves from transgenic plants were frozen in liquid nitrogen and ground into fine powder with a Tissuelyser (Qiagen). Soluble protein was extracted in 1 mL of 50 mM phosphate buffer (pH 7.0) supplemented with 0.1% [v/v] Triton X-100 and cOmplete™ Protease Inhibitor Cocktail (Roche, 11873580001, one tablet per 100 mL). Protein concentration then determined using a Qubit protein assay kit (Invitrogen, Q33212). The MUG fluorescent assay was performed in duplicates with 20 µl protein extract in MUG assay buffer (50 mM phosphate buffer (pH 7.0), 10 mM EDTA-$Na_2$, 0.1% [v/v] Triton X-100 (Sigma, X100), 0.1% [w/v] N-lauroylsarcosine sodium (Fluka analytical, 61747), 10 mM DTT (Melford, D11000), 2 mM 4-methylumbelliferyl-β-D-glucuronide (4-MUG, Sigma, M9130)) in a 200 µl total volume. The reaction was conducted at 37 °C in GREINER 96 F-BOTTOM microtiter plate using a CLARIOstar plate reader. 4-Methylumbelliferone (4-MU) fluorescence was recorded every 2 min for 20 cycles with excitation at 360 nm and emission detected at 450 nm. 4-MU concentration was determined based on a standard curve of ten 4-MU standards placed in the same plate. GUS enzymatic rates were calculated by averaging the slope of MU production from each of the duplicate reactions.

In order to visualize mTurquoise2 signal, mature leaves were dissected into 2 cm sections, leaf epidermal cells were removed by scraping the leaf surface with a razor blade and then mounted with deionized water. Imaging was then performed using a Leica TCS SP8 confocal laser-scanning microscope using a 20x air objective. mTurquoise2 fluorescence was excited at 442 nm with emission at 471–481 nm, chlorophyll autofluorescence was excited at 488 nm with emission at 672–692 nm.

## Yeast one hybrid, protoplast isolation and transactivation assay

The yeast one hybridisation assay was performed by Hybrigenics (https://www.hybrigenics-services.com/). DNA fragments were synthesised and used as bait, rice leaf and root cDNA libraries were used as prey. The number of clones screened and concentration of 3-AT were

as follows: fragment 1, 70.2 million clones screened with 0 mM 3-AT; fragment 2, 61.5 million clones screened with 0 mM 3-AT; fragment 3, 68.4 million clones screened with 20 mM 3-AT; fragment 4, 57.4 million clones screened with 100 mM 3-AT; fragment 5, 94.2 million clones screened with 200 mM 3-AT.

For transactivation effector assays, transcription factors were cloned using primers list in Supplementary Fig. 13 based on a number of selection criteria: first, transcription factors from families that were preferentially expressed in the bundle sheath of mature leaves (belonging to BS or BSV clusters in Hua et al., 2021[30]) including transcription factors that were preferentially expressed in BSS during leaf maturation (co-expression modules 15&17 in the leaf developmental gradient dataset (Supplementary Data 5)), and transcription factors that were co-expressed with *SiR* (module 0047) were identified from a publicly available co-expression network RiceGGM2021[124]. Additionally, other members of the IDD and group-A bZIP families were cloned (Supplementary Fig. 13). Coding sequences of these candidate transcription factors were domesticated as described[116] and amplified using primers listed in Supplementary Data 9 and prepared as level 0 SC module in the background of *pAGM9121*. They were then assembled into a level 1 module with a *ZmUBIpro* promoter and *Tnos* terminator as effector plasmids. Nucleotides −980 to −829 with the endogenous core promoter (nucleotide −250 to +42) were fused with the *LUCIFERASE* (*LUC*) reporter. All golden gate level 1 modules for protoplast transfection were extracted using ZymoPURE™ II Plasmid Midiprep Kit (Zymo research, D4201).

Rice leaf protoplasts and PEG-mediated transformation were performed as described previously[125]. In each transformation, 2 µg of transformation control plasmids (*ZmUBIpro::GUS-Tnos*), 5 µg of reporter plasmids, and 5 µg of effector plasmids per transcription factor were combined and mixed with 170 µl protoplasts. After incubation on the benchtop for overnight, protein was extracted using passive lysis buffer (Promega, E1941), GUS activity was determined with 20 µl of protein sample and MUG fluorescent assay as described above, LUC activity was measured with 20 µl of protein sample and 100 µl of LUC assay reagent (Promega, E1483) using Clariostar plate reader. Transcriptional activity from the promoter was calculated as LUC luminescence / rate of MUG accumulation per second.

## Reporting summary

Further information on research design is available in the Nature Portfolio Reporting Summary linked to this article.

# Data availability

RNA sequencing data for bundle sheath strands and mesophyll from mature rice leaf (IR64) have been deposited in the Sequence Read Archive (SRA) under accession PRJNA1205909. RNA sequencing data of bundle sheath strands and mesophyll during leaf developmental gradient are available under accession PRJNA1205924. Sequences of *ZmPEPC* promoter (PQ873046 [https://www.ncbi.nlm.nih.gov/nuccore/PQ873046.1/]), *SiR* promoter (PQ873047 [https://www.ncbi.nlm.nih.gov/nuccore/PQ873047]), the bundle sheath CRM (PQ873048 [https://www.ncbi.nlm.nih.gov/nuccore/PQ873048]), three copies of the bundle sheath CRM (PQ873049 [https://www.ncbi.nlm.nih.gov/nuccore/PQ873049]), five copies of the bundle sheath CRM (PQ873050 [https://www.ncbi.nlm.nih.gov/nuccore/PQ873050]), *SiR* core promoter (PQ873051 [https://www.ncbi.nlm.nih.gov/nuccore/PQ873051]) and *PIP1.1* core promoter (PQ873052 [https://www.ncbi.nlm.nih.gov/nuccore/PQ873052]) are deposited in GenBank. Source data are provided with this paper.

# Code availability

Scripts used for searching Y-patch and the motif sextet can be accessed at https://github.com/hibberd-lab/rice-bundle-sheath-cis-regulatory-module (https://doi.org/10.5281/zenodo.15681256).

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

## Acknowledgements

This work was funded by the Bill and Melinda Gates Foundation C₄ Rice grant awarded to the University of Oxford (2015-2019 OPP1129902 and 2019-2024 INV-002970), L.H., N.W., S.S., R.D., K.B., S.K.E., A.R.B., and J.M.H. for the purpose of open access, the authors have applied a Creative Commons Attribution (CC BY) licence to any Author Accepted Manuscript version arising from this submission.

## Author contributions

L.H. and J.M.H. conceived the work. J.M.H. guided execution of experiments and oversaw the project. L.H., N.W., S.S., R.D., K.B., S.K.E. and A.R.B. did the experiments and analysed the data. L.H. and J.M.H. wrote the manuscript with input from all authors.

## Competing interests

The authors declare no competing interests.
