## [Transparent Peer Review file · Nature Communications]

A transcription factor ensemble orchestrating bundle sheath expression in rice

Corresponding Author: Professor Julian Hibberd

Version 0:

Reviewer comments:

Reviewer #1

(Remarks to the Author)

Hua et al identified the detailed cis-regulatory elements of the SULFITE REDUCTASE (SiR) promoters responsible for their bundle sheath cell specificity, based on a systematic truncation of SiR promoters. Additionally, TFs from five TF families associated with the core cis-regulatory region, or enhancer region, were proposed and identified. The functions of TF binding sites for these TFs families were also revealed using truncation analysis. The authors demonstrated that this mechanism is conserved in the dicot species Arabidopsis, providing evidence of a convergent mechanism between dicot and monocot species. In general, the study used robust experimental methods to verify their results, such as RNA-seq to validate predicted transcription start sites (TSS) and misexpression bZIP9, IDD2, and WRKY121 in mesophyll cells (using the ZmPEPC promoter) to support that BSC specificity was a result of the SiR enhancer rather than the specific expression of BSC TFs. The identifying specific promoter regions and transcription factors that drive BSC specificity in rice is novel and valuable. However, there are several unclear points and missing information in the methods section, and many long sentences make the text difficult to understand. The author should address these issues to improve clarity and completeness.

Major comments:

1. The authors selected 7 genes most strongly expressed in bundle sheath strands (BSS) and showed their expression levels in MC (mesophyll cells) and BSS. However, the details of how LCM was performed and how the expression levels were measured are not provided. I assume RNA-seq was performed, as the expression levels are measured in TPM. If this is the case, RNA-seq data and quantification methods should be provided.
2. Why was SiR not detected during the initial LCM method but was detected in their early study (Hua et al. 2021)? Since SiR does not show expression in MC but exhibits BS specificity, I wonder if the BS specificity of SiR only occurs in early stages or certain stages. The developmental stage of GUS measurement should be specified in the methods or figure legends.

Minor comments

1. Line 22-23: "We identified the SULFITE REDUCTASE promoter as sufficient for strong bundle sheath expression." Change "as" to "was."
2. Line 63-66: Two sentences are mixed. "In leaves of these angiosperms, particular cell types are specialized for photosynthesis and so whilst photosynthesis gene expression is induced by light in all major cell types of the rice leaf the response is greater in spongy and palisade mesophyll cells compared with guard, mesophyll and bundle sheath cells." These should be separated into two independent sentences.
3. Line 91-92: Specify the exact region of the promoter sequence fused to the GUS reporter gene.
4. Line 158: How were the cis-regulatory elements predicted?
5. Line 220-221: "When consecutive deletions to this sequence were made, statistically significant reductions in MUG activity were evident but there was no impact on accumulation of GUS in the bundle sheath." This result should reference the corresponding figure or table.
6. Line 241, 241. "873" should be "-873".
7. Lines 532: What transformation control plasmids were used in the transactivation assay?
8. Supplementary Figure 3: Provide information on the number of replicates.
9. Supplementary Figure 6B: Clarify the specific regions of promoter sequences from the compared species.

10. The detailed information of ZmPEPC used in this study was not provided.

11. Except for the SiR promoter, the primers for amplifying other promoters or cloning the coding sequences (CDS) of studied TFs were not provided.

Reviewer #2

(Remarks to the Author)

In this manuscript, the authors characterized the SiR promoter which they show drives bundle sheath specific expression in rice. They used a series of promoter truncations and fusions along with TF cis-motif analysis and other approaches to identify regions that were important for this bundle sheath specificity. Based on their characterization of the motif, they also designed synthetic promoters that enabled high and specific expression in the rice bundle sheath and show this construct also drives bundle sheath expression in Arabidopsis.

Strengths: The work addresses the important question of what determines cell type specificity in the leaf and has significant implications for crop engineering efforts. Ultimately, the authors show that they can use the knowledge from their characterization to engineer a synthetic promoter that drives strong bundle sheath specific expression in rice and Arabidopsis

Weaknesses/ aspects to address:

1) It was not clear in several instances what criteria were used to follow specific lines of inquiry, and therefore it is hard for me to judge how to interpret the relevance/robustness of the results. For example, how were the TFs that were tested in Fig 4B, 4C, 4E and S14 selected? E.g. given that there are ~100 WRKY TFs, why were only those three tested? Also notable that there is no overlap with Y1H TFs. Likewise, the initial 25 promoters tested were genes "...more abundant in bundle sheath cells compare with veins and mesophyll..." from Hua et al, but it isn't clear how that was determined and how comprehensive this set is. Given that there are several published methods for cell specificity, e.g. tau index, a description of what was done here and the cutoffs used would help ensure that this analysis was robust. And finally, how were the "promoter" regions from these genes selected, at least one goes well downstream of translation start site and lengths vary from ~1500bp to >3kb?

2) The identification and engineering of the SiR promoter to generate strong BS specific expression is impressive, however the analysis and conclusions on TF-DNA interactions could be developed further to broaden the impact on our understanding of the regulatory logic that drives this cell specificity. For instance, there appears to be an interaction between the overlapping regions specifying BS expression (-980 to -829) and mesophyll suppression (-980/-829 to -700) which leads to the conflicting results seen when comparing the deletions/truncations in the "full" fragment compared to enhancer + core fragments in Figs S5, 3E, S10 and S11 (e.g. different effects of removing b or c from each). Another example regards the Y-patch, which does appear to be important, but I have concerns about the conclusions that are drawn about "correlation" of expression to number/length from using only PIP1;1, NRT1.1A and SiR, when it is only 3 core promoters and those were genes that had BS enriched expression based on the LCM data. To make such claims would require additional core promoters (ideally series of synthetic promoters with different numbers of Y-patches) and the appropriate controls. Lastly, the authors generalize the importance of these binding sites by pointing to their cooccurrence in other BS enriched genes, however there is no analysis to say whether this is unusual when looking at a what is essentially a 5000bp window around the TSS or whether these other clusters are functional.

3) Related to the previous two points, there are several places where my initial interpretation is that 5 specific TFs that control BS expression have been identified, most notably the title, whereas ultimately the results implicate promoter regions containing TF cis-motifs for which a subset of candidates were tested from relevant TF families. This discrepancy leads to odd phrasing by the authors themselves, e.g. on lines 96-97 with where specific TFs vs TF families are conflated.

4) I think it would be appropriate to edit or remove entirely the paragraph on Lines 327-332. The data that supports this is from Hua et al, and those genes were also discussed in that work. As it is worded, it gives the sense that the BS expression of sulphur metabolism genes here was independent evidence for that finding, whereas in my opinion it is validation of those findings.

Minor points:

Lines 101-128 - Figure S1A seems unnecessary as it could be combined with S2, and the source of the data is not clear (citation?). Seems the message being conveyed is that it is important to compare M, BS and V expression, but can just state this more clearly and concisely.

Fig S2A - Add vein expression data. Would also be useful to have genes in A/B/C be in same order

Lines 120-121 - Text states SiR promoter drives GUS expression only in BS but Fig1A/B, S2 A/C, and Hua et al indicate there is some expression in veins as well.

Fig S5 – Based on comparison to Fig 2, there appears to be an error in the activity plot with -829 to +42 being switched with the plot for -980 to -251/min35s.

Line 179/224– "abolished GUS accumulation" – Is it truly abolished? What is the activity in non-transgenic (or control) lines as a reference?

Fig 3 – Consistency with other figures. Panel A should probably show the construct schematic as it isn't as clear as in other figures what was fused to these

Line 234 – Should “Historically” be “experimentally”?

Line 264-265 and 346 – Looks to be additive effects rather than synergistic.

Line 337 – suggest replacing “here” with “in that study” or similar as I first thought you were talking about this study.

Line 375 – Fairly sure this should say -980 to -700 based on Fig S11 and region specified in Fig 6.

What is the source of the cell-specific leaf developmental gradient dataset (needs citation)?

Version 1:

Reviewer comments:

Reviewer #1

(Remarks to the Author)

I am pleased to note that the authors have thoroughly addressed all of my previous concerns. They have provided the RNA-seq data as requested and offered a clear explanation for their choice of SiR in this study. Furthermore, they have justified the selection of SiR and verified its expression patterns across different developmental stages through supplementary experiments. The additional information on RNA sequencing data and quantification methods has added substantial value to the study. I appreciate the authors' careful attention to the feedback and their diligent efforts in revising the manuscript.

Overall, the authors have made significant enhancements to the rigor and completeness of the paper, addressing all raised issues effectively. Therefore, I recommend this manuscript for acceptance in Nature Communications.

Reviewer #2

(Remarks to the Author)

The authors have done an excellent job clarifying their approach by providing more detail on the experimental methods used and links to the underlying data. The additional assay of synthetic Y-patch constructs and enrichment analysis are quite convincing. They have addressed all of the major issues that I identified in the initial submission.

Personally, I think "team" sounds odd and would go with something like "ensemble," but that is obviously up to them.

RESPONSES TO REVIEWERS' COMMENTS

Reviewer #1 (Remarks to the Author):

Hua et al identified the detailed cis-regulatory elements of the SULFITE REDUCTASE (SiR) promoters responsible for their bundle sheath cell specificity, based on a systematic truncation of SiR promoters. Additionally, TFs from five TF families associated with the core cis-regulatory region, or enhancer region, were proposed and identified. The functions of TF binding sites for these TFs families were also revealed using truncation analysis. The authors demonstrated that this mechanism is conserved in the dicot species *Arabidopsis*, providing evidence of a convergent mechanism between dicot and monocot species. In general, the study used robust experimental methods to verify their results, such as RNA-seq to validate predicted transcription start sites (TSS) and misexpression bZIP9, IDD2, and WRKY121 in mesophyll cells (using the ZmPEPC promoter) to support that BSC specificity was a result of the SiR enhancer rather than the specific expression of BSC TFs. The identifying specific promoter regions and transcription factors that drive BSC specificity in rice is novel and valuable.

Response: *Thank you for this very positive comment of our work.*

However, there are several unclear points and missing information in the methods section, and many long sentences make the text difficult to understand. The author should address these issues to improve clarity and completeness.

Response: *We have addressed these points below.*

Major comments:

1. The authors selected 7 genes most strongly expressed in bundle sheath strands (BSS) and showed their expression levels in MC (mesophyll cells) and BSS. However, the details of how LCM was performed and how the expression levels were measured are not provided. I assume RNA-seq was performed, as the expression levels are measured in TPM. If this is the case, RNA-seq data and quantification methods should be provided.

Response: *Thank you for this point. We confirm that RNA sequencing was performed using RNA isolated from bundle sheath strands and mesophyll cells from rice mature leaf using laser capture microdissection. More detailed information about the LCM protocol and RNA-seq method is now included in the Materials and Methods section (lines 474–489). Additionally, the differential gene expression analysis have been added to Supplemental Table 3. The full RNA-seq dataset is publicly available at BioProject accession PRJNA1205909.*

2. Why was SiR not detected during the initial LCM method but was detected in their early study (Hua et al. 2021)? Since SiR does not show expression in MC but exhibits BS specificity, I wonder if the BS specificity of SiR only occurs in early stages or certain stages. The developmental stage of GUS measurement should be specified in the methods or figure legends.

Response: *Thank you for your comment. We apologise that these data were not presented in a clear enough manner. In the first iteration, where we dissected bundle sheath cells together with veins from mature 4th leaves, SiR was indeed detected as preferentially expressed in bundle sheath strands among 791 other genes (Log2 fold change > 1 and adjusted P < 0.05 from analysis using DESeq2) (Supplemental Table 3). But, we did not select the SiR promoter to test because we chose genes that were more differentially and highly expressed, however it turned out that these genes were primarily expressed in veins.*

Subsequently, we separated bundle sheath from veins using LCM and RNA-seq using Kitaake mature leaves and SiR was then identified as highly expressed in bundle sheath and veins. At this point, we chose SiR promoter to test its specificity. We have modified the text at lines 103-104 by highlighting that bundle sheath strands contain veinal cells, we also modified Figure S2 panel A to include the vein data so that expression of SiR is clearer.

To test the expression pattern of SiR promoter during leaf development, we performed GUS staining of primordia at plastochron P3 whilst leaf 4 was 5 mm to 8 cm long. Our data

indicate that SiR is first detected at P3, and this pattern is maintained during leaf maturation, it is also detected in mature leaves. This is consistent with data from our developmental gradient LCM RNA-seq analysis where SiR transcripts were more highly expressed in bundle sheath strands than in mesophyll cells across the 17 days of leaf developmental gradient (Supplemental Figure 4Q). The developmental stage of GUS measurement measured by leaf 4 length has been specified in figure legends of Supplemental Figure 4.

Minor comments

1. Line 22-23: "We identified the SULFITE REDUCTASE promoter as sufficient for strong bundle sheath expression." Change "as" to "was."

Response: We apologise that this sentence lacked clarity. We have modified it such that it now reads "Using the global staple and C₃ crop rice we found that the SULFITE REDUCTASE promoter was sufficient for strong bundle sheath expression".

2. Line 63-66: Two sentences are mixed. "In leaves of these angiosperms, particular cell types are specialized for photosynthesis and so whilst photosynthesis gene expression is induced by light in all major cell types of the rice leaf the response is greater in spongy and palisade mesophyll cells compared with guard, mestome and bundle sheath cells." These should be separated into two independent sentences.

Response: We agree that the original text was clumsy. In fact, we realised that some of the information was not needed to get the key message across, and so we contracted what could have been two sentences into one. The text now reads "In leaves of these angiosperms, particular cell types are specialised for photosynthesis because their response to light is greater". Thank you for this point.

3. Line 91-92: Specify the exact region of the promoter sequence fused to the GUS reporter gene.

Response: We have added the exact region of the promoter relative to translational start site in the text Line 91 to 92.

4. Line 158: How were the cis-regulatory elements predicted?

Response: Thank you for your comment. Cis-regulatory elements were predicted using the FIMO program with JASPAR motif position weight matrix, we have added the information in the text at lines 164-165, and for detailed information please see "Motif analysis" section in Materials and Methods at lines 561-570.

5. Line 220-221: "When consecutive deletions to this sequence were made, statistically significant reductions in MUG activity were evident but there was no impact on accumulation of GUS in the bundle sheath." This result should reference the corresponding figure or table.

Response: Thank you for spotting this, the figure reference has been added in line 228.

6. Line 241, 241. "873" should be "-873".

Response: Thank you for spotting this. It has been corrected – now at line 248.

7. Lines 532: What transformation control plasmids were used in the transactivation assay?

Response: Thank you for this point. The ZmUBIpro::GUS-Tnos construct was used as the transformation control, we have modified the methods section to include this information at lines 639-640.

8. Supplementary Figure 3: Provide information on the number of replicates.

Response: Thank you, number of replicates has now been added in the plot.

9. Supplementary Figure 6B: Clarify the specific regions of promoter sequences from the compared species.

Response: Thank you, regions have now been added.

10. The detailed information of ZmPEPC used in this study was not provided.

Response: Thank you for this point, the references have been added at line 553. The Genebank ID for this promoter can be found in Data Availability section.

11. Except for the SiR promoter, the primers for amplifying other promoters or cloning the coding sequences (CDS) of studied TFs were not provided.

Response: We apologise for this omission. Primer sequences for cloning promoters and transcription factors have been provided as Supplemental Table 7, 8 and 9 respectively.

Reviewer #2 (Remarks to the Author):

In this manuscript, the authors characterized the SiR promoter which they show drives bundle sheath specific expression in rice. They used a series of promoter truncations and fusions along with TF cis-motif analysis and other approaches to identify regions that were important for this bundle sheath specificity. Based on their characterization of the motif, they also designed synthetic promoters that enabled high and specific expression in the rice bundle sheath and show this construct also drives bundle sheath expression in Arabidopsis.

Strengths: The work addresses the important question of what determines cell type specificity in the leaf and has significant implications for crop engineering efforts. Ultimately, the authors show that they can use the knowledge from their characterization to engineer a synthetic promoter that drives strong bundle sheath specific expression in rice and Arabidopsis.

Response: Thank you for this very positive analysis of our work.

Weaknesses/ aspects to address:

1) It was not clear in several instances what criteria were used to follow specific lines of inquiry, and therefore it is hard for me to judge how to interpret the relevance/robustness of the results. For example, how were the TFs that were tested in Fig 4B, 4C, 4E and S14 selected? E.g. given that there are ~100 WRKY TFs, why were only those three tested?

Response: Thank you for this point. We apologize for the lack of clarity regarding the selection criteria for transcription factors tested in the effector assay. The transcription factors were chosen based on multiple criteria, with a primary focus on their cell specific co-expression with SiR (LOC_Os05g42350). We also cloned close relatives to assess redundancy. Specifically, we used the following filters to identify candidates:

- 1. Initially we cloned transcription factors preferentially expressed in the bundle sheath (BS) or BS vascular (BSV) tissues, and/or those co-expressed with SiR in the bundle sheath strand (BSS) during leaf development (module 15&17). Examples include WRKY121, IDD2, bZIP9, and bZIP10.*
- 2. Secondly, trans-factors co-expressed with SiR in the co-expression module M0047, identified through the publicly available RiceGGM2021 co-expression network (Zhang et al., 2022) were also included. Examples include WRKY1, SNAC1 (NAC9), and NAC3, 5, and 6.*
- 3. Lastly, direct orthologs of JASPAR transcription factors were selected. Additionally, members of the IDD and group-A bZIP families, were cloned to test redundancy.*

We have clarified these criteria in the text (lines 266-269) and in method (lines 623-631). Detailed information for each transcription factor that met these criteria was provided in Supplemental Figure 13. The co-expression analysis during leaf maturing has been provided as Supplemental table 5&6.

Also notable that there is no overlap with Y1H TFs.

Response: This is a good point, this is likely because yeast one-hybrid assay is designed to detect interactions between a single transcription factor and DNA sequence and is less suited for scenarios where multiple transcription factors need to form a complex to activate transcription (Sewell & Fuxman Bass, 2018). This is likely the case for the bundle sheath CRM, and so we now make this point in the Discussion in line 361-364.

Likewise, the initial 25 promoters tested were genes "...more abundant in bundle sheath cells compare with veins and mesophyll..." from Hua et al, but it isn't clear how that was determined and how comprehensive this set is. Given that there are several published methods for cell specificity, e.g. tau index, a description of what was done here and the cutoffs used would help ensure that this analysis was robust. And finally, how were the "promoter" regions from these genes selected, at least one goes well downstream of translation start site and lengths vary from ~1500bp to >3kb?

Response: Thank you for raising these points, The initial 25 promoters tested were selected based on their higher expression in bundle sheath and vein or bundle sheath alone compared to the mesophyll, as identified in Hua et al. 2021. The tau index is less easy to use with our data as it is based on whether a gene is cell specific relative to all other cell types, whereas the genes we were interested are preferential to both bundle sheath and vein, or preferential to the bundle sheath alone. We therefore selected genes belonging to bundle sheath or bundle sheath and vein co-expression clusters (Hua et al., 2021), and further narrowed down candidates by differential expression analysis with DESeq2 and edgeR using the following criteria: $\text{Log}_2\text{FC}(\text{BS}/\text{M}) > 2$ & DESeq2 adjusted P (BS/M) < 0.01 & edgeR FDR (BS/M) < 0.01 and $\text{Log}_2\text{fc}(\text{BS}/\text{V}) > -0.5$. 210 genes matched these criteria. We then chose genes associated with biological processes enriched in the bundle sheath, such as solute transport, sulfur metabolism, and nitrogen metabolism. We have now included a detailed description of this analysis in Materials and Methods at lines 494-498, as well as text at lines 117 to 120). The full list of BS-specific genes is now provided in Supplemental Table 7.

Regarding the selection of "promoter" regions - these were designated as a minimum of 1500-bp upstream sequence from the translational start site or entire intergenic region if shorter than 1500-bp. In some cases, this was extended both upstream and downstream of the TSS to capture regulatory elements identified through DNase hypersensitive sites (DHS) at the Plant DHSs Database (<https://plantdhs.org/>; Zhang et al., 2016b) We have clarified this selection strategy and its rationale in the revised text (lines 104-106) and in the method section (lines 522-526).

2) The identification and engineering of the SiR promoter to generate strong BS specific expression is impressive, however the analysis and conclusions on TF-DNA interactions could be developed further to broaden the impact on our understanding of the regulatory logic that drives this cell specificity. For instance, there appears to be an interaction between the overlapping regions specifying BS expression (-980 to -829) and mesophyll suppression (-980/-829 to -700) which leads to the conflicting results seen when comparing the deletions/truncations in the "full" fragment compared to enhancer + core fragments in Figs S5, 3E, S10 and S11 (e.g. different effects of removing b or c from each).

Response: Thank you for the comment. We agree that further exploration of TF-DNA interactions would enhance our understanding of the regulatory logic underlying BS specificity. Regarding the interaction between the bundle sheath enhancer (-980 to -829) and mesophyll repression (-829 to -700), we do not understand the exact mechanism underlying this interaction, but our data predicts that it might be mediated by transcription factors preferentially expressed in the mesophyll as well as so far unknown factors that binds nucleotides -829 to -700. We have updated the working model in Figure 6E and indicate this in the text at lines 392-395.

Another example regards the Y-patch, which does appear to be important, but I have concerns about the conclusions that are drawn about "correlation" of expression to number/length from using only PIP1;1, NRT1.1A and SiR, when it is only 3 core promoters and those were genes

that had BS enriched expression based on the LCM data. To make such claims would require additional core promoters (ideally series of synthetic promoters with different numbers of Y-patches) and the appropriate controls. Lastly, the authors generalize the importance of these binding sites by pointing to their cooccurrence in other BS enriched genes, however there is no analysis to say whether this is unusual when looking at a what is essentially a 5000bp window around the TSS or whether these other clusters are functional.

Response: Thank you for this point – we agree that our data were limited here. To address whether copy or length of Y-patch correlate with gene expression level, we performed the effector assay using synthetic core promoters with two or three copies of the Y-patch and compared this to one copy as a control. In addition, we also made longer Y-patches of 40-60 bp to compare with the endogenous 20 bp sequence. This showed that more copies of Y-patches enhanced transcriptional output by *GLK2*, *MYBS1*, *IDD10* and *bZIP*, but that the degree of response was most striking for *IDD10*. A longer pyrimidine rich sequence did not have this effect. These data are now included in Supplemental Figure 17 and summarised in the text lines 383-385.

In order to assess the general importance of the bundle sheath CRM and Y-patch, we searched for Y-patch in core promoter and sextet of motifs in open chromatin region from promoter and gene body within 300-bp window with a 100-bp step in rice genome, and found additional 282 genes also containing Y-patch and the bundle sheath CRM, of these, 107 are expressed in bundle sheath (TPM>5) and 21 are preferential to the bundle sheath, fisher's exact test showed that genes containing this module in bundle sheath are over-represented over genes not containing this module, indicating this module is important for bundle sheath expression in general, we also found five other genes involved in sulphur metabolism including *Fd*, *CLT1*, *STR22* as well as two transcription factors *EIL2* & *3* considered as master regulators for sulphur metabolism and ethylene signalling, hence this module is particularly important for sulphur metabolism. We have modified the text in lines 364-374 and relevant method in lines 572-585, we also provided the list of genes in Supplemental table 10.

3) Related to the previous two points, there are several places where my initial interpretation is that 5 specific TFs that control BS expression have been identified, most notably the title, whereas ultimately the results implicate promoter regions containing TF cis-motifs for which a subset of candidates were tested from relevant TF families. This discrepancy leads to odd phrasing by the authors themselves, e.g. on lines 96-97 with where specific TFs vs TF families are conflated.

Response: Thank you. This is a good point, and we admit we were struggling on how to phrase this because these transcription factors are encoded by multigene families, and multiple members of each family activate the enhancer. We now opted to use the term transcription factor team to introduce the work and provide when we need to provide an overview of what is going on. When assessing the details of which transcription factors bind this CRM, we refer to both the gene families, and the particular genes involved. Please see the modified title and text in line 25-26, 97-98, 302, 304.

4) I think it would be appropriate to edit or remove entirely the paragraph on Lines 327-332. The data that supports this is from Hua et al, and those genes were also discussed in that work. As it is worded, it gives the sense that the BS expression of sulphur metabolism genes here was independent evidence for that finding, whereas in my opinion it is validation of those findings.

Response: Thank you for this point and we agree with you that the conclusion is redundant with Hua et al., 2021. The text has been removed, and we modified surrounding text to ensure the narrative is clearly focussed on the role of upstream regulatory regions that are important for sulphur metabolism. Please see lines 340-342.

Minor points:

Lines 101-128 - Figure S1A seems unnecessary as it could be combined with S2, and the source of the data is not clear (citation?). Seems the message being conveyed is that it is

important to compare M, BS and V expression, but can just state this more clearly and concisely.

Response: Thank you for the point, as the source of data as well as the method for generating the data are different in these two figures, we have opted to keep them separate. However, we recognise the point that you are making and so have modified the relevant text. In Figure S1A, bundle sheath cells were sampled together with veins, but this led to identification of genes that are more highly expressed in veins. We therefore performed another LCM RNAseq where we separated bundle sheath from veins. We now have clarified these points by adding reference to bundle sheath strands as bundle sheath together with the vascular bundle including xylem, phloem in the text at lines 103-104.

Fig S2A - Add vein expression data. Would also be useful to have genes in A/B/C be in same order

Response: Thank you, we have modified the figure and table.

Lines 120-121 - Text states SiR promoter drives GUS expression only in BS but Fig1A/B, S2 A/C, and Hua et al indicate there is some expression in veins as well.

Response: This is a good point, our initial text was sloppy on this, we have revised the text in now at line 126.

Fig S5 – Based on comparison to Fig 2, there appears to be an error in the activity plot with -829 to +42 being switched with the plot for -980 to -251/min35s.

Response: Thank you for spotting this, Figure 2 has been revised and is now correct.

Line 179/224– “abolished GUS accumulation” – Is it truly abolished? What is the activity in non-transgenic (or control) lines as a reference?

Response: Thank you for this point. We have revised the text to read “diminished”.

Fig 3 – Consistency with other figures. Panel A should probably show the construct schematic as it isn't as clear as in other figures what was fused to these.

Response: We have revised panel A accordingly.

Line 234 – Should “Historically” be “experimentally”?

Response: Yes, we have altered this to experimentally in line 241.

Line 264-265 and 346 – Looks to be additive effects rather than synergistic.

Response: Thank you, the log-transformed data do appear to be additive in the graph, however, the scale is logarithmic, and absolute values for the LUC/GUS data indicate that co-expression of GLK2 with the IDD3 leads to a 2-3 fold increase compared with the two transcription factors combined. These data therefore do support a synergistic rather than additive effect of the transcription factors. We now also present the raw data in Supplemental figure 14F.

Line 337 – suggest replacing “here” with “in that study” or similar as I first thought you were talking about this study.

Response: We agree with this point, and the text now reads “In that study...” in line 347.

Line 375 – Fairly sure this should say -980 to -700 based on Fig S11 and region specified in Fig 6.

Response: Thank you for this point, we have modified the text in lines 388-390.

What is the source of the cell-specific leaf developmental gradient dataset (needs citation)?

Response: Raw reads of the cell-specific leaf developmental gradient RNA-seq data are now publicly available under accession PRJNA1205924. We are currently finalising a separate manuscript that reports a separate analysis of these data.